# Neuron hemilineages provide the functional ground plan for the *Drosophila* ventral nervous system

Robin M Harris†, Barret D Pfeiffer, Gerald M Rubin, James W Truman*

Janelia Research Campus, Howard Hughes Medical Institute, Ashburn, United States

**Abstract** *Drosophila* central neurons arise from neuroblasts that generate neurons in a pair-wise fashion, with the two daughters providing the basis for distinct A and B hemilineage groups. 33 postembryonically-born hemilineages contribute over 90% of the neurons in each thoracic hemisegment. We devised genetic approaches to define the anatomy of most of these hemilineages and to assessed their functional roles using the heat-sensitive channel dTRPA1. The simplest hemilineages contained local interneurons and their activation caused tonic or phasic leg movements lacking interlimb coordination. The next level was hemilineages of similar projection cells that drove intersegmentally coordinated behaviors such as walking. The highest level involved hemilineages whose activation elicited complex behaviors such as takeoff. These activation phenotypes indicate that the hemilineages vary in their behavioral roles with some contributing to local networks for sensorimotor processing and others having higher order functions of coordinating these local networks into complex behavior.

## Introduction

A major obstacle to understanding how the central nervous system (CNS) translates sensory inputs into appropriate motor outputs is the CNS's staggering cellular complexity. Even the small CNS of *Drosophila* has on the order of 100,000 neurons (*Power, 1943*; *Chiang et al., 2011*), each of which is capable of making connections with dozens of synaptic partners. To understand such a complex system, it is helpful to parse its elements into relevant units organized in a functional hierarchy. This approach can be seen in the analysis of the vertebrate brainstem and spinal cord, which are the major sites of sensorimotor processing for locomotor behaviors. In the spinal cord, for example, identified progenitor cells generate discrete pools of interneurons, which in turn have defined roles in producing functional circuits (*Grillner and Jessell, 2009*). The functional organization of the spinal cord in vertebrates appears to be conserved, in that the progenitor cells and transcription factors that characterize different interneuron classes are conserved from fish through mammals (*Lupo et al., 2006*) despite the marked differences in modes of locomotion. The hindbrain of the zebra fish is similarly structured from clusters of cell types that are recruited in predictable ways to assemble functional circuits (*Koyama et al., 2011*).

Like the vertebrate spinal cord, the ventral nervous system (VNS) of insects is the site of sensorimotor patterning for complex behaviors such as walking, jumping, and flight. Years of work on grasshoppers and other orthopteroid insects (summarized in *Burrows, 1996*) have defined clusters of spiking and nonspiking interneurons that transform sensory input into changes in leg position during the maintenance of stance (*Burrows, 1996*) and organize more complex behaviors such as walking (*Buschges et al., 2008*) and flight (*Robertson et al., 1982*). As in vertebrates, these clusters follow a developmental logic in which specific clusters arise from the same neuronal stem cell (neuroblast [NB]). In the grasshopper, for example, NB 4-1 has been shown to produce the cluster of midline

*For correspondence: trumanj@janelia.hhmi.org

†Deceased

**eLife digest** The legs and wings of insects are borne on the middle body segments, which make up the thorax. The nervous system inside of the thorax is part of the insect equivalent of the spinal cord and contains clusters of interneurons that relay signals between the sensory nerves, the brain and the muscles. This enables the insect to perform complex actions such as walking and flying.

The thoracic interneurons are produced by a fixed set of stem cells. Each stem cell makes neurons in a pair-wise fashion by producing a sequence of neural progenitor cells, each of which then divides to produce two different types of daughter neurons. All of the daughter neurons of the same type are said to belong to the same hemilineage, and in the fruit fly *Drosophila*, the majority of the interneurons in the thorax are from one of 33 hemilineages. Each interneuron cluster in the insect thorax is made up of cells from a single hemilineage.

Harris et al. developed genetic tools that allow the different hemilineages in the *Drosophila* thorax to be labeled, and used this to create a set of flies that allows the role of the different clusters to be investigated. Each fly type was modified so that increasing the temperature activated a heat-sensitive channel in the neurons of a single hemilineage, and Harris et al. recorded the behavioral response this produced.

Each hemilineage caused the fly to move in a distinctive way when stimulated, and many of these movements were unique to a single cluster. Furthermore, the hemilineages can be divided into different groups based on their complexity. Activating the simplest group of hemilineage clusters produces simple movements such as leg twitches and stretches. Another group of hemilineages are then able to organize these movements into more complicated behaviors, such as walking. The third, most complex, hemilineages can coordinate several complex actions to enable the flies to perform very complicated tasks, like take off for flight.

These findings suggest that hemilineages act as the basic modules of the nervous system in the fly thorax. Furthermore, the flies and techniques developed by Harris et al. will provide valuable resources for future studies into the organization and function of the nervous system.

spiking interneurons that receive exteroceptor input and shape the receptive fields of leg motoneurons (*Shepherd and Laurent, 1992*). Similarly, the unpaired medial NB produces the pool of local, midline GABAergic neurons that respond to sound (*Thompson and Siegler, 1991*).

The finding that specific stem cells generate pools of functionally similar interneurons has important evolutionary implications because the insect VNS develops according to a conservative plan based on a segmental set of 30 paired and one unpaired NBs that has changed little through the 350 million years of insect evolution (*Thomas et al., 1984*; *Truman and Ball, 1998*). Each NB is uniquely identifiable and characterized by its position in the array, its pattern of molecular expression (*Broadus and Doe, 1995*), and the suite of early neurons that it produces (*Bossing et al., 1996*; *Schmid et al., 1999*). Comparative studies between a basal, primitively flightless insect and flying insects showed that the number of thoracic neurons roughly doubled with the evolution of flight. This increase did not come from adding new NBs, but rather was correlated with a subset of the NBs making more neurons by extending their proliferative phase (*Truman and Ball, 1998*). This mixed proliferative response associated with the evolution of flight is consistent with the idea that particular neuronal lineages have become adapted to specific behavioral functions, but this has yet to be comprehensively evaluated.

The systematic decoding of the behavioral roles of the segmental set of NBs can best be carried out in *Drosophila* because of its life history and the genetic tools available. Its metamorphic life history is advantageous because the great majority of central neurons arise during a second neurogenic period as the larva grows. During this postembryonic phase, each NB divides repeatedly, with the smaller product of each division being a ganglion mother cell (GMC), which then terminally divides to produce two neurons. By the start of metamorphosis each NB and its immature progeny form a discrete cell cluster from which one or two axon bundles project to specific regions in the larval neuropil (*Truman et al., 2004*). Each axon bundle identifies the neurons of a hemilineage, a set of neuronal 'cousins' that includes either the 'A' (Notch-on) or 'B' (Notch-off) daughter from the GMC division (*Truman et al., 2010*). In *Drosophila*, 25 of the 30 embryonic NBs in a hemisegment generate postembryonic lineage. Of the 50 possible hemilineages, 17 are removed by programmed cell death

so that a segmental unit of the adult thoracic CNS contains neurons from only 33 hemilineage-based pools. These account for 90–95% of the neurons in the adult thoracic CNS. Hemilineages are also units for the early molecular diversity within the VNS. Expression of many early transcription factors is restricted along hemilineage lines (*Lacin et al., 2014*) and many of these transcription factors have homologs involved in fate determination in the vertebrate spinal cord (*Thor and Thomas, 1997*). For example, the homeodomain protein Dbx1 controls the difference between the V0 and V1 fates in the mouse spinal cord (*Pierani et al., 2001*), while the *Drosophila* homolog, *dbx*, controls differentiation of particular VNS cell types through interactions with *even-skipped* and *hb9* (*Lacin et al., 2009*).

We developed a set of genetic tools to allow us to examine the form and function of most of the thoracic hemilineages in *Drosophila*. We find that most hemilineages are composed of pools of similar interneurons, confirming that neuronal classes are indeed based on a hemilineage plan. Stimulation of interneurons in a hemilineage pool using the temperature-sensitive cation channel dTRPA1 (*Hamada et al., 2008*) elicits a characteristic, often unique, behavioral response from each. The complexity of the elicited behavior reflects the complexity of the hemilineage's projection pattern and the neuropil region to which it projects. These findings suggest that the hemilineages provide a functional as well as an anatomical ground plan for the thoracic nervous system, and that thorax-mediated behaviors occur by combinations of simple movements elicited by relatively simple ventral hemilineages, which are then orchestrated by a hierarchy of increasingly complex dorsal hemilineages.

## Results

### Strategies for achieving hemilineage expression

The neurons in a hemilineage are designated by their NB of origin and whether they are the Notch-on (A) or Notch-off (B) daughters of the GMC division (*Truman et al., 2010*). For example, hemilineages 1A and 1B are the respective A and B daughters of NB 1. We are using the postembryonic designations of the NBs (from *Truman et al., 2004*) rather than their embryonic names (i.e., *Schmid et al., 1999*) because there is controversy over the correspondence of the two maps (*Birkholz et al., 2015*; Lacin and Truman, in preparation). Working with isolated hemilineages has been difficult because the members are cousins rather than sisters. Consequently, a clonal approach, such as MARCM (*Lee and Luo, 1999*; *Yu et al., 2010a*), serves to mark the 2 daughters of the GMC division, but does not label all the neurons in one hemilineage at the exclusion of those in the other. Therefore, we devised approaches to mark the secondary neurons in a hemilineage and then track them through metamorphosis to assess their form and function in the adult. Of the 33 major thoracic hemilineages, hemilineages 0A, 4A, 16B and 17A are not addressed in this study.

Since each hemilineage can be identified in the larva based on cluster position and trajectory of its axon bundle (*Truman et al., 2010*), we screened the larval expression patterns of the Rubin GAL4 collection (*Jenett et al., 2012*) based on over 7000 cis-regulatory regions (CRMs; see *Pfeiffer et al., 2008*), for lines that drove expression in single hemilineages (*Li et al., 2014*). The larval pattern was then maintained by using a transiently expressed recombinase to remove a transcriptional stop cassette from an Actin5C -FRT>-stop-FRT>-LexA::p65 transgene and thereby extending expression into the adult stage. The successful application of this strategy, though, required that the recombinase activity was limited to the larval growth period when the desired hemilineage pattern was expressed.

Because of the diversity of expression patterns in the driver lines, we explored three different methods for implementing this strategy (summarized in *Figure 1*). The results of the first method (see *Figure 1A*) is illustrated in *Figure 2*. In the last instar larva, line R24B02-GAL4 drives expression in the cells of hemilineage 12A from the subesophageal through the A1 segments (*Figure 2A*). However, this expression pattern wanes during metamorphosis and is replaced by a different set of cells in the adult CNS (*Figure 2B*). To maintain lineage expression into the adult, we used R24B02-GAL4 to drive pJFRC180-20XUAS-IVS-Flp2::PEST, hereafter referred to as UAS-Flippase, to remove the transcriptional stop from an Actin5C-FRT>-dSTOP-FRT>-GAL4 construct. As shown in *Figure 2C*, when combined with the reporter construct, pJFRC2-10XUAS-IVS-mCD8::GFP (*Pfeiffer et al., 2010*), expression in the hemilineage 12A cells was maintained into the adult. A complication, though, was that expression was observed in other cells, presumably ones in which the stop cassette had been removed in earlier larval or embryonic stages, when this CRM drives a different expression pattern. To circumvent this early expression we used a GeneSwitch strategy (*Osterwalder et al., 2001*; *Roman et al., 2001*) in which we fused a *Drosophila* codon-optimized ligand-binding domain of the human

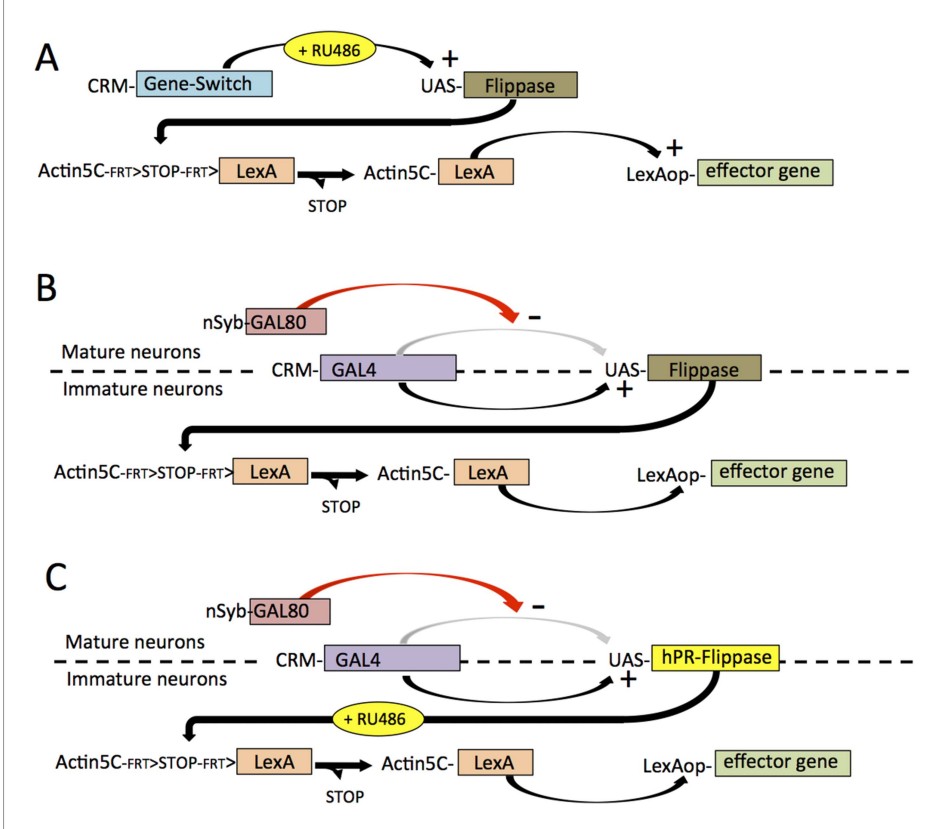

**Figure 1**. The different strategies that were used to establish lines that showed selective expression in defined hemilineages. The strategies are based on a screen through the large collection of enhancer lines built from *cis* regulatory modules (CRMs) of CNS expressed genes. (**A**) For CRMs whose thoracic expression is confined to a hemilineage, gene-switch constructs are combined with feeding larvae the progesterone mimic (RU486) in the last larval stage. The larval expression of flippase then promotes the excision of a STOP cassette from another trans-gene allowing a constitutive promotor (Actin5C) to drive continual expression following excision. (**B**) When the larval expression pattern includes functional larval neurons as well as a hemilineage, expression in the larval neurons is blocked by including a nSynaptobrevin-GAL80 gene. Gene switch cannot be used in this context because it is not suppressed by GAL80. (**C**) Spatial and temporal specificity is accomplished using a conditional flippase that is the human progesterone receptor ligand-binding domain (hPR) fused to Flippase. Exposure of third instar larvae to RU486 then confines the flip event to the last larval stage. See text for more details.

progesterone receptor to the DNA-binding domain of GAL4. R24B02-GeneSwitch should then drive expression only after larvae are treated with a progesterone mimic, such as RU486. As seen in *Figure 2D,E* feeding larvae RU486 during the third instar resulted in strong expression of immature 12A interneurons. Removal of the drug at the start of metamorphosis was followed by a waning of GFP expression so that it was lost by adult emergence (*Harris, 2012*). Feeding adults with RU486 reinduced GFP expression but only in the expected adult pattern and not in the 12A interneurons. Consequently, the RU486 given to feeding larvae is effectively cleared from the animal during metamorphosis (this study). We then used the R24B02-GeneSwitch line in conjunction with UAS-flippase, Actin5C-FRT>-dSTOP-FRT>-GAL4, and pJFRC2-10XUAS-IVS-mCD8::GFP to reveal the adult morphology of the hemilineage 12A neurons. Without treatment with RU486 in the larva, we saw no GFP expression at any stage (*Figure 2D,F*), but feeding them with the drug during the third larval instar resulted in expression in the 12A interneurons that persisted through metamorphosis and continued to be robust in the adult (*Figure 2G,H*). Importantly, this adult expression pattern did not also contain the adult-specific cells characteristic of R24B02 because the RU486 was cleared from the system before these cells started to express late in metamorphosis.

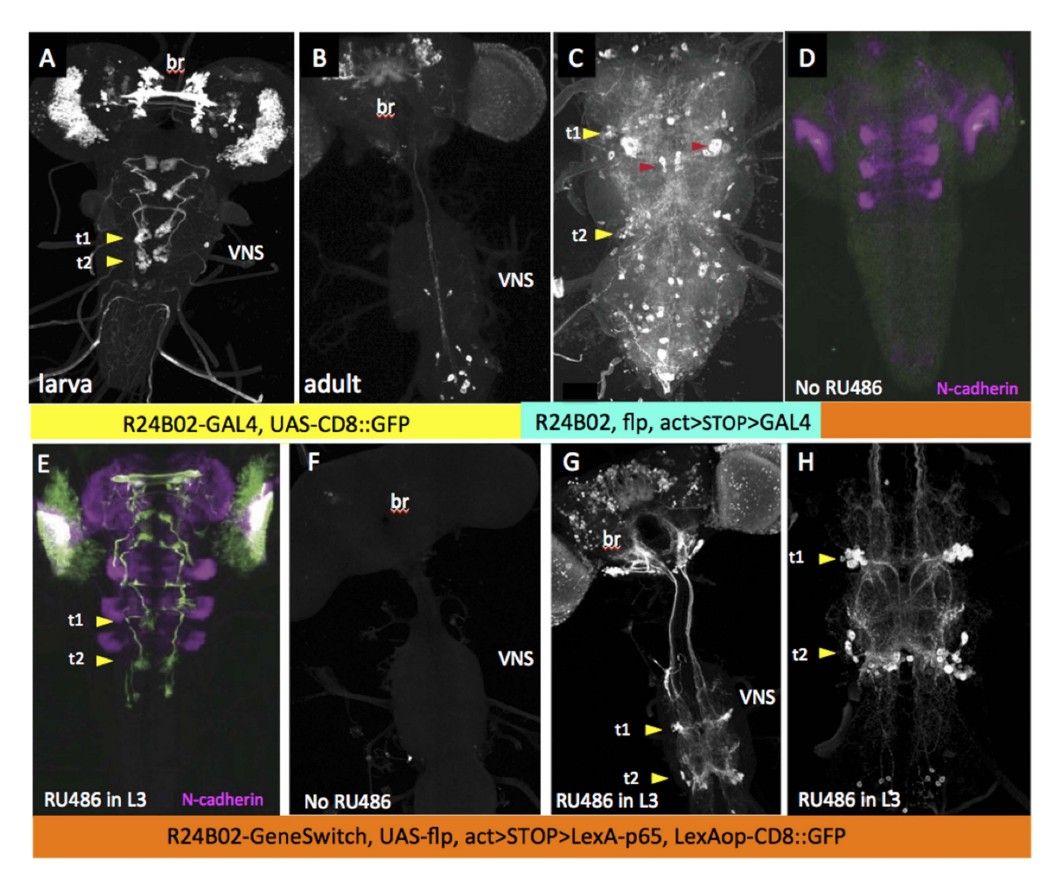

**Figure 2**. Strategy for clean expression of hemilineages into the adult stage. Z-projections of confocal stacks showing the expression pattern driven by R24B02 used in various genetic combinations. Arrowheads show the t1 and t2 clusters of hemilineage 12A. (**A**, **B**) Pattern shown by R24B02-GAL4 driving pJFRC2-10XUAS-IVS-mCD8::GFP (pJFRC2) in larval (**A**) and adult (**B**) stages. The hemilineage 12A clusters are prominent at the end of larval life, but do not express in the adult. (**C**) Adult VNS of a cross of Actin5C>dSTOP>GAL4, UAS-Flippase; pJFRC2 to R24B02-GAL4. The persisting expression in the hemilineage 12A clusters is badly obscured by many 'off-target' cells (e.g., red arrowheads) presumably arising from embryonic and adult expression patterns in this line. (**D**, **E**) R24B02-GeneSwitch flies crossed to pJFRC2 and either maintained without hormone (**D**) or fed on 1 mM RU486 food for 24 hr as third instar larvae. (**F**–**H**) Adult nervous systems of flies of the genotype R24B02-GeneSwitch, Actin5C>dSTOP>GAL4 , UAS-Flippase, pJFRC2 and either raised without hormone mimic (**F**) or fed RU486 food during the third larval stage (**G**, **H**). H shows a higher power view of the hemilineage 12A cells found in T1, T2 and A1; this hemilineage dies in T3. Green: GFP; magenta: N-cadherin.

A second strategy (see *Figure 1B*) to remove extraneous expression from hemilineage lines was based on the fact that the arrested secondary neurons do not express terminal differentiation products such as synaptic vesicle proteins. We first confirmed that a driver for such a gene—R57C10-GAL4 that carries an 872-bp promoter fragment from the nSynaptobrevin gene (nSyb; *Pfeiffer et al., 2008*; *Pfeiffer et al., 2012*)—drives expression in the functioning primary neurons, but not in the clusters of arrested immature secondary neurons (*Figure 3A*, inset). This promoter fragment was then used to drive expression of GAL80, an inhibitor of GAL4 activity (*Yun et al., 1991*; *Traven et al., 2006*), in the construct R57C10-GAL80-6, hereafter called nSyb-GAL80. When crossed into lines that contained a mixture of functioning larval neurons and clusters of arrested, postembryonic cells (*Figure 3B*), nSyb-GAL80 suppressed expression in the mature neurons, leaving only the expression in the arrested immature neurons (*Figure 3C*). When nSyb-GAL80 was used in conjunction with a GAL4 diver, UAS-flippase and Actin5C-FRT>-dSTOP-FRT>-LexAp::65 , we were able to fix—or 'immortal-ize'—the expression pattern of that GAL4 pattern specifically in the immature neurons, now as

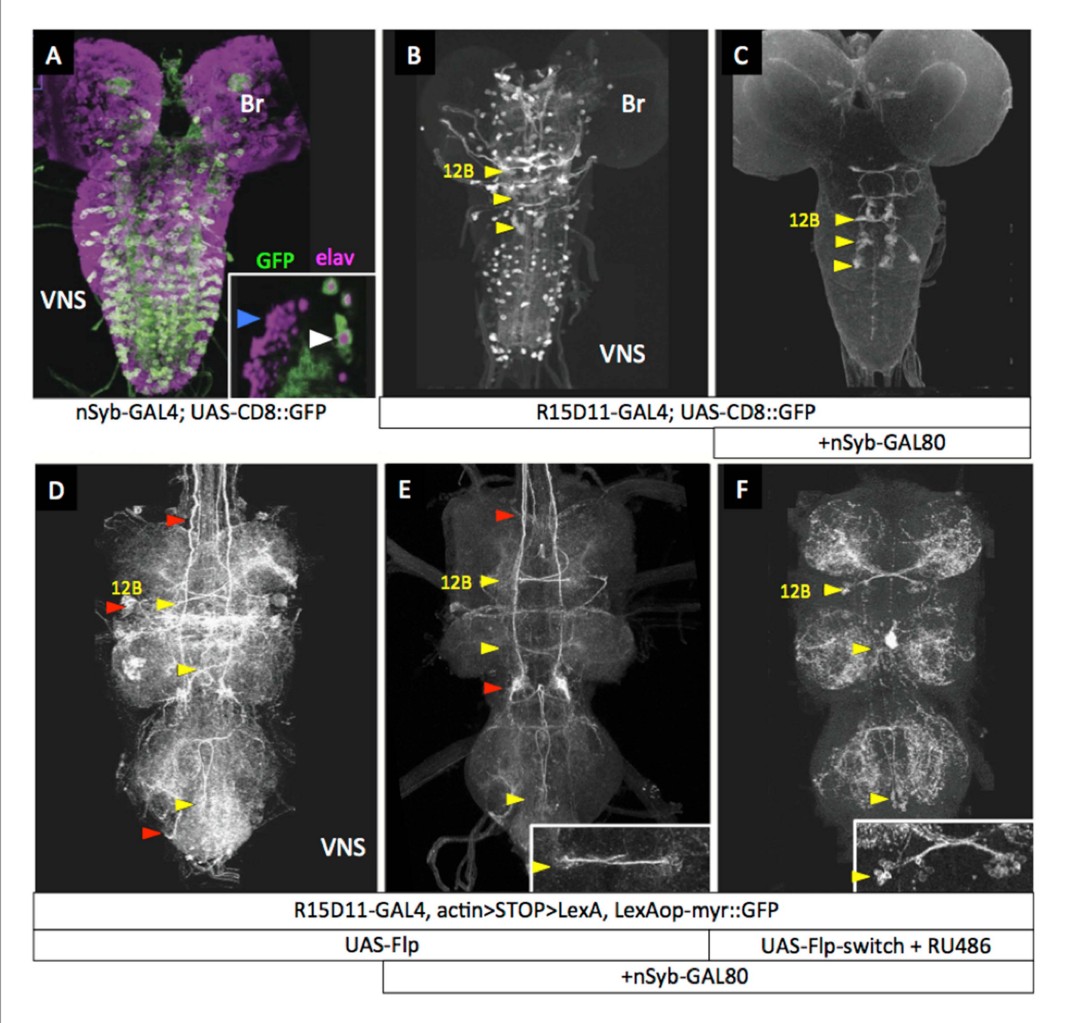

**Figure 3**. nSyb-GAL80 suppresses GAL4 expression specifically in mature neurons. Yellow arrowheads show location of hemilineage 12B cell clusters in the thoracic segments. (**A**) In the third-instar larva, R57C10-GAL4, a nSynaptobrevin promoter fusion-GAL4 drives expression in primary neurons, but not secondary neurons. Inset: single optical slice showing colocalization of anti-GFP (green) and the pan-neural marker anti-elav (magenta) in primary neurons (e.g., white arrowhead), but not in immature secondary neurons (e.g., blue arrowhead). (**B**) R15D11-GAL4 drives expression in secondary hemilineage 12B (yellow arrowheads) and various primary neurons. (**C**) R15D11-GAL4 with nSyb-GAL80: expression is suppressed in primary neurons, and only 12B expression remains. (**D**) When used to drive UAS-Flippase in a flip-on immortalization strategy, R15D11-GAL4 yields expression in hemilineage 12B, but also in numerous off-target cells (red arrowheads). Genotype: w; UAS-Flippase (attP40)/pJFRC19-13XLexAop2-IVS-myr::GFP (attP40), Actin5Cp4.6>dsFRT>LexAp65 (su(Hw)attP5); R15D11-GAL4 (attP2). (**E**) Same genotype as (**D**), but with nSyb-GAL80 on the X chromosome (su(Hw)attP8). Expression in off-target cells is much reduced. A small amount of off-target expression is observed in secondary neurons and descending neurons (e.g., red arrowheads). (**F**) Adult VNC expression pattern using both nSyb-GAL80 and UAS-Flp-Switch to restrict expression to the targeted cells in (**C**). No off-target expression is observed. In panels **D–F**, myr::GFP concentrates in processes rather than in the cell bodies and therefore the cell clusters are difficult to see in Z-projections. **E**, **F**: insets are partial projections showing t1 cell body clusters.

The following figure supplement is available for figure 3:

**Figure supplement 1**. UAS-hPR-flp is fully active in the presence of RU486, but inactive without drug.

a pattern of LexA activity, which was observed using pJFRC19-13XLexAop2-IVS-myr::GFP (*Pfeiffer et al., 2010*). However, significant off target expression was observed (*Figure 3D,E*) because we did not have temporal control over the timing of the flippase activity.

To make a more precise tool (depicted in *Figure 1C*), we fused the codon-optimized ligand-binding domain of the human progesterone receptor to the Flp recombinase, making recombinase activity dependent on the presence of RU486 (*Figure 3F*). *Figure 3* (*Figure 3—figure supplement 1*) shows a test of this construct, pJFRC108-20XUAS-IVS-hPR::Flp-p10, which we call UAS-Flp-Switch. Expression of GFP in larval nervous systems carrying R20B05-GAL4, pJFRC177-10XUAS-FRT>-dSTOP-FRT>-myr::GFP (*Nern et al., 2011*), and the UAS-Flp-Switch showed very strong expression that was conditional on feeding larvae on RU486-containing food during the third larval instar (*Harris, 2012*). Using the Flp-Switch system in conjunction with GAL80 suppression (*Figure 1C*), we could then obtain clean expression in hemilineage 12B cells in the adult (*Figure 3F*).

By tracking hemilineages through metamorphosis, we were able to define which neuroglian-positive tracts were used by each hemilineage in the adult (*Harris, 2012*). With this knowledge, we rescreened the GAL4 collection and in a few instances found a line that cleanly showed expression in an adult hemilineage. We then used a combination of these few lines as well as ones that were generated using a combination of the nSyb-GAL80 and Flp-Switch approaches to obtain coverage for most of the thoracic hemilineages. The genotypes selected for accessing each of the hemilineages in the adult are listed in *Supplementary file 1*, which also assesses off-target expression in the lines.

## Association of hemilineage anatomy with function in the adult VNS

Each thoracic hemilineage has a unique projection pattern that allows it to be distinguished from the others. The only exceptions are hemilineages 20A and 22A, which are from neighboring NBs. These two hemilineages cannot be readily distinguished in the larva (*Truman et al., 2004*) and we were unable to differentiate them in our analysis. Consequently, they are considered as a single unit. An important diagnostic feature of each hemilineage is the neuroglian-positive tracts in which it runs. A detailed analysis of the larval-to-adult transformation of this tract system is described in Shepherd et al. (2015, in revision), so only a superficial treatment will be given here. The immortalized driver lines show the neuropil regions that are targets of a given hemilineage, but for the more complex hemilineages, that overlap either bilaterally or intersegmentally, a detailed analysis of the anatomy needs to be supplemented by a clonal approach that has been used sparingly here and will be dealt with in detail elsewhere (Shepherd, Sahota, Court, Harris, Truman and Williams, in preparation). The thoracic neuropil can be roughly divided into the paired, ventrolateral leg neuropils in each segment and a dorsal region, the tectulum (*Power, 1943*), which has lost obvious segmental organization and is most expanded in the T2 region.

To understand the behavioral function of each hemilineage, we activated the neurons in decapitated flies and observed their response. Decapitated flies maintain a good stance and show little spontaneous leg or wing movements except for occasional grooming bouts of the front legs or of the hind legs over the wings (*Video 1*). Cells in the respective hemilineages were excited using the temperature-sensitive TRPA1 channel expressed in the corresponding neurons and the decapitated flies subjected to a linear heat ramp and videos were acquired through the warming process for 45 to 55 s (see 'Materials and methods'). Decapitated flies lacking the TRPA1 effector show no response to the heat ramp, maintaining their posture with occasional bouts of grooming movements. The behavioral responses of the decapitated flies were divided into six behavioral categories. (1) *Changes in posture*: this category includes tonic changes in posture or in leg or joint position. (2) *Uncoordinated leg movements*: the flies showed phasic leg movements that had no obvious inter- or intrasegmental coordination. The decapitated flies typically stayed in place or, if they moved, their course was erratic. (3) *Walking*: this involved translocation of the animal over more than a body length during the test period. The

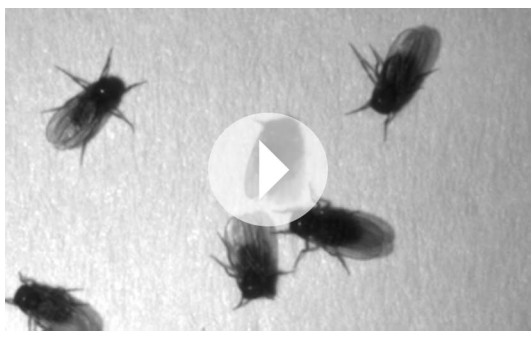

**Video 1.** Control, decapitated flies of the genotype R24B02-GAL4 subjected to a 24˚ to 37˚C heat ramp and recorded at 60 frames per second (fps). This video related to *Figure 4*.

movements could be forwards, sideways or backwards, but the trajectories were typically smooth. (4) *Wing waving*: this category includes bouts of low frequency wing movements such as wing extension, wing flicking, or wing scissoring. (5) *Wing buzzing:* these movements include high frequency wing movements that appeared as a blur on the 60 fps video frames. The wings could be extended laterally in the flight or singing positions or could remain over the back. During buzz episodes the decapitated fly might or might not become airborne. (6) *Takeoff*: these typically involved the decapitated fly launching into the air within one to two frames of the normal video (span of ~30–60 ms). In the lines in which this response was common, we used high-speed video to confirmed the role of the t2 legs in producing the takeoff jump.

## Hemilineages 1A and 1B

The lineage 1 cluster in the larva is located in the anterolateral region of the segment and has a 1A bundle that projects in the ventral anterior (vA) commissure to the contralateral leg neuropil and a 1B bundle that projects to the leg neuropil in the next anterior segment. During metamorphosis the somata of the 1A and 1B clusters are pulled apart so that the 1B somata become situated at the posterior edge of the next anterior segment (*Figure 4E*). The 1B interneurons are local interneurons that have arbors in the ventromedial and dorsolateral regions of the leg neuropil (*Figure 4F*). The adult 1A interneurons, by contrast, are projection neurons that travel through the vA commissure and form one of the most ventral tracts in the VNS (*Figure 4A,B*). After crossing the midline, the 1A bundle makes a distinctive posterior 'hook' (*Figure 4A*) and then bifurcates into dorsal and ventrolateral projections (*Figure 4B*, *Figure 4—figure supplement 1*). The ipsilateral and contralateral arbors project into similar neuropil areas so that the dorsal and ventral arbors from the right and left clusters appear to overlap. A synaptotagmin::GFP fusion protein (nSyt::GFP) (*Zhang et al., 2002*; *Pfeiffer et al., 2010*; *Seelig and Jayaraman, 2013*) localizes primarily in the ventral arbors (*Figure 4C*) in the intermediate layers of the leg neuropil, showing the primary output regions of these cells.

Since our best 1B line showed expression in a dorsal hemilineage in addition to the 1B neurons, we did not use it for behavioral observations. Activation of 1A interneurons via TRPA1 evoked forward locomotion. As the temperature ramped up, the decapitated flies started to walk, with the total distance covered ranging from just over a body length to greater than ten lengths during the trial period (*Figure 4D*; *Video 2*). Locomotion was also occasionally interrupted by bouts of grooming. The behavior displayed by these flies was similar to that described as locomotion for decapitated flies exposed to biogenic amines in *Yellman et al. (1997)*. During locomotion the leg movements were erratic and not organized in the tripod gait typical of walking in intact flies (*Strauss and Heisenberg, 1990*), but there was clear intersegmental coordination of the limbs. The net movement is always forwards, although the trajectory turns due to uneven step sizes, as observed by *Yellman et al. (1997)*. Rarely, a fly would takeoff during the heat ramp.

## Hemilineage 2A

The 2A interneurons in the larva represent the surviving hemilineage from lineage 2. They are situated as paired clusters on either side of the midline at the anterior margin of each thoracic neuromere. Their axons project dorsally and then spread laterally over the ipsilateral dorsal neuropil. This anatomy is conserved in the adult, with the 2A interneurons extending broadly within the ipsilateral part of the tectulum (*Figure 4G–I*). Arbor from all three segments show a degree of convergence into the T2 region.

Activation of the 2A interneurons drove high frequency wing movements (54%, n = 25). The decapitated flies typically stood in place but as the temperature increased, they abruptly initiated high frequency flapping with the wings extended laterally in the flight position (*Figure 4J*; *Video 3*). This wing buzzing was usually maintained for the remainder of the heat ramp, but it was occasionally interrupted by a bout of wing grooming. A few of the buzzing flies eventually went airborne. We do not know if this was accomplished by a jump via their T2 legs or if their tarsi simply lost contact with the substrat.

## Hemilineages 3A and 3B

In the larva, the lineage 3 interneurons are in a ventromedial cluster at the posterior border of the segment. The axons of the A and B daughters project dorsally to the mid-neuropil, where the 3A

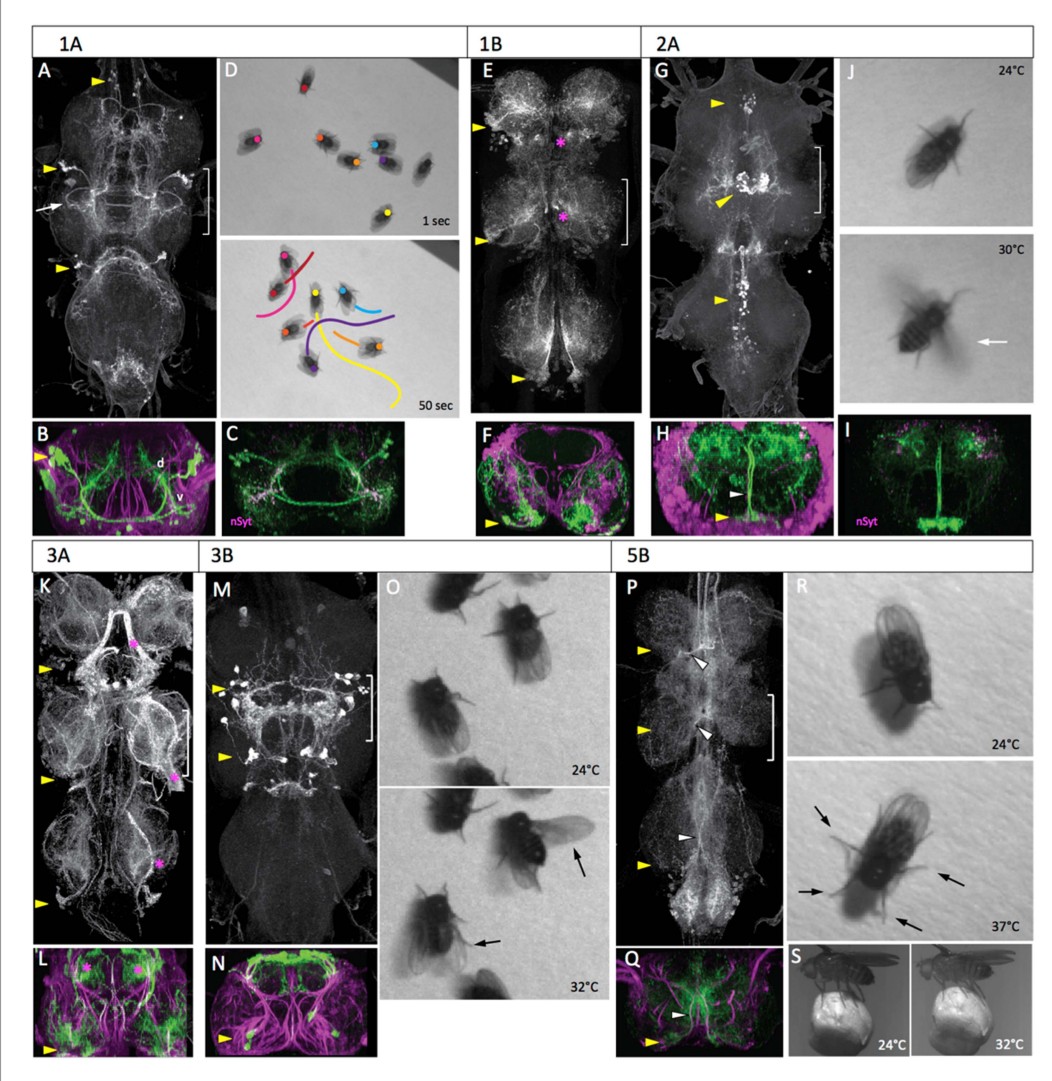

**Figure 4**. The anatomy and behavioral consequences of stimulating hemilineages 1A through 5B. Each hemilineage is depicted as a projected confocal Z-stack of the VNS (T1, T2, T3 and the fused abdominal neuromeres) (**A**, **E**, **G**, **K**, **M**, **P**) and a transverse projection through segment T2 (**B**, **F**, **H**, **L**, **N**, **Q**; bracketed region in dorsal view). yellow arrowheads: hemilineage cell body clusters; white arrowheads: main neurite bundle entering neuropil or crossing midline; magenta asterisk: major off-target expression; green: GFP; magenta: neuroglian. Pictures are video frames of groups of decapitated flies that express TRPA1 in the particular hemilineage and are exposed to a heat ramp to stimulate the neurons. (**A–D**) Hemilineage 1A. (**A**, **B**) The 1A neurons are located dorsolaterally in each thoracic segment, project across the midline and have ventral (v) arbor in the leg neuropil and dorsal (d) arbor in the tectulum neuropil. Arrow: characteristic posterior hook of the ventral arbor. (**C**) T2 transverse projection showing that nSynaptotagmin::GFP (nSyt, magenta) localizes to the ventral arbor. (**D**) The response of activating the 1A neurons in decapitated flies with a 24–37°C heat ramp over a 50 s period. Images show the position of marked flies at the beginning and end of the ramp and the path each moved during the period. (**E**, **F**) Hemilineage 1B. The 1B neurons are located in the posterior vertrolateral region of each segment and send arbors in to ventral and dorsal regions of the ipsilateral leg neuropil. (**G–J**) Hemilineage 2A. (**G**, **H**) The 2A neurons are situated ventromedially in the anterior third of the ganglion; they project dorsally and arborize throughout the ipsilateral tectulum. (**I**) T2 transverse projection showing that nSyt-GFP (magenta) localizes in the more lateral parts of the arbor. (**J**) Activation of the 2A neurons by the head ramp results in buzzing of the outstretched wings (30°C; arrow). (**K**, **L**) Hemilineage 3A. The 3A interneurons are in a posterior ventrolateral cluster; they enter the leg neuropil near the leg nerve and ramify through most of the ventral half of the leg neuropil. The line had substantial sensory expression (*). (**M–O**) Hemilineage 3B. (**M**, **N**) The 3B interneuron clusters are in posterior T1 and T2 and project dorsally to ramify through the dorsal part of the tectulum. (**O**) Thermal activation of the 3A primarily evokes flicking and scissoring movements

*Figure 4. continued on next page*

*Figure 4. Continued*

of the wings (arrows). (**P–S**) Hemilineage 5B. (**P**, **Q**) The 5B clusters are positioned ventrolaterally in the posterior part of each thoracic neuromere. They project dorsally and across the intermediate posterior commissure to arborize in dorsal and medial regions of the neuropil. (**R**, **S**) Thermal activation of the 5B interneurons cause decapitated flies to splay out their legs (**R**, 37°C) and tethered intact flies to crouch onto a Styrofoam ball they are holding (**S**, 32°C). Genotypes: for most genotypes see *Supplementary file 1*; for the remainder: (**C**) LexAop-RFP (su(Hw)attP8)/w; R22G11-LexA (attP40)/+; LexAop-nSyt-GFP (su(Hw)attP1)/+. (**I**) UAS-nSyt-GFP (attP18)/w; +; R50G08-GAL4 (attP2)/ UAS-HA (VK00005).

The following figure supplement is available for figure 4:

**Figure supplement 1**. Dorsal (**A**) and transverse (**B**) view of the adult VNS showing a MARCM clone for the T2 lineage 1.

neurons enter the ipsilateral leg neuropil, while the 3B daughters continue into the ipsilateral dorsal neuropil. In the adult, the cell bodies of hemilineage 3A remain ventral, just posterior to the leg nerve insertion and their arbors extend through intermediate levels of the leg neuropil (*Figure 4K,L*). The cluster of 3B interneurons is severely reduced or absent in T3. In T1 and T2 the arbors of the cells converge in the tectulum to form a complex arbor (*Figure 4M,N*) in the very dorsal layer of this neuropil.

The 3A line had major contamination from sensory neurons and was not used for behavioral observations. The effects of activating the 3B interneurons were seen using the R23B05-GAL4 line driving TRPA1 (*Figure 4O*, *Video 4*). As the temperature ramped up, the decapitated flies showed a subtle repositioning of the legs, with some repetitive, poorly coordinated leg movements and occasional grooming bouts. They also performed occasional wing flicking and wing scissoring movements but did not show high frequency wing buzzing.

## Hemilineage 5B

In the larva, hemilineage 5A undergoes programmed cell death (*Truman et al., 2010*) leaving only the 5B interneurons. The hemilineage 5B somata are ventrolateral, and their neurites project through the intermediate posterior (iP) commissure and arrest in an ascending tract. In the adult, the cell bodies are pulled towards the midline. Their bundled axons cross the midline in the iP commissure, and form a major ipsilateral arbor just prior to crossing (*Figure 4P,Q*). The contralateral arbors of the cells make a compact projection that extends up and down the medial region of the neuropil and into the neck connective.

When the 5B interneurons were activated, the decapitated flies showed a progressive repositioning of their limbs so that they are closer to the ground by the end of the thermal ramp (*Figure 4R*, *Video 5*). Spontaneous grooming appeared to be suppressed and we observed no walking or wing associated behaviors. Tethered intact flies that were gripping a Styrofoam ball responded to the temperature ramp by lowering their body position, thereby drawing the ball closer to them (*Figure 4S*).

## Hemilineages 6A and 6B

In the larva, both lineage 6 hemilineages project across the midline with the 6A bundle forming the dorsal posterior (dP) commissure and 6B bundle crossing in the iP commissure. During metamorphosis the cell bodies of the two

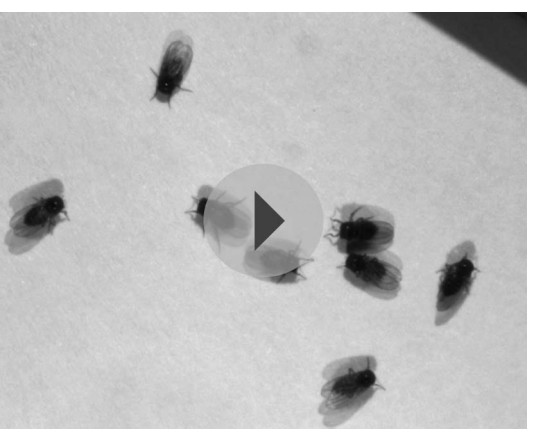

**Video 2.** Behavioral effects of exciting the neurons in hemilineage 1A. This video related to *Figure 4*. Decapitated flies expressing TRPA1 under control of R22G11 are subjected to a 24° to 37°C heat ramp and recorded at 60 fps. Full genotype in *Supplementary file 1*.

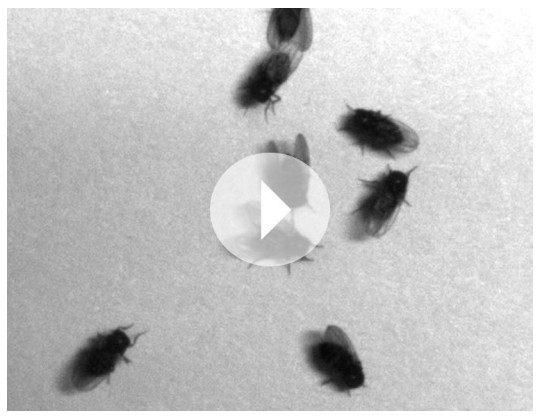

**Video 4.** Behavioral effects of exciting the neurons in hemilineage 3B. This video related to *Figure 4*. De-capitated flies expressing TRPA1 under control of R23B05 are subjected to a 24° to 37°C heat ramp and recorded at 60 fps. Full genotype in *Supplementary file 1*.

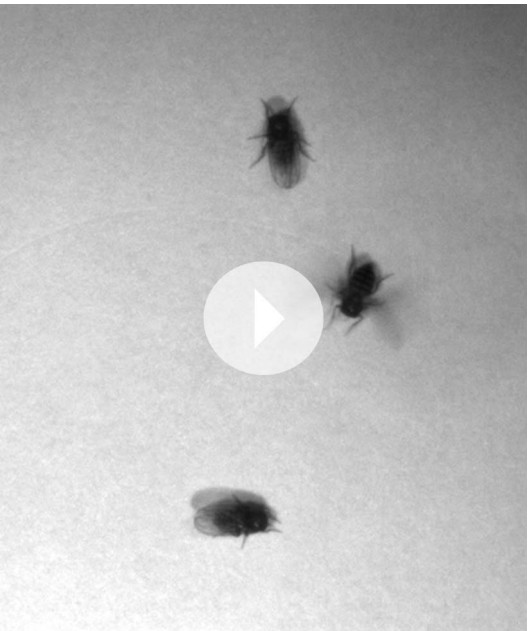

**Video 3.** Behavioral effects of exciting the neurons in hemilineage 2A. This video related to *Figure 4*. De-capitated flies expressing TRPA1 under control of R50G08 are subjected to a 24° to 37°C heat ramp and recorded at 60 fps. Full genotype in *Supplementary file 1*.

hemilineages separate into distinct clusters, but stay ventral (*Figure 5B,F*), as described in *Brown and Truman (2009)*. Hemilineage 6A clusters are found in neuromeres S3 through A1 (*Figure 5A*) and form one of the dorsalmost arbors of the tectulum. The 6A clusters show an ipsilateral arbor that is largely confined to the segment of origin and a distal, contralateral arbor that converges from S3 through A1 into the T2 region of the tectulum (*Figure 5A,B*). nSyt::GFP localizes to the lateral portions of the contralateral arbor (*Figure 5C*). The neurons of 6B cluster are found in T1 to T3 but typically are missing from segment A1. The 6B bundle crosses in the iP commissure without an ipsilateral arbor and then branches profusely in the tectulum neuropil and the dorsal-most region of the leg neuropils (*Figure 5E,F*). As with the 6A neurons, the T1 and T3 clusters tend to converge onto T2.

Activation of the 6A interneurons via the heat ramp started with poorly coordinated movements of the legs that often pitched the fly forward onto its anterior thorax (*Figure 6D*, *Video 6*). The movements were not organized into any particular direction and became very erratic towards the end (*Figure 6D*, 30-31s). The decapitated flies occasionally showed wing flicking and high frequency buzzing of the wings but the wings were usually only partially spread rather than in the extended flight position.

Activation of the 6B interneurons in the decapitated flies elicited a mixture of leg-related movements (*Figure 5G*; *Video 7*). Some showed a curving, forward movement while others pivoted in place. Most flies did not move far enough (>one body length) to be classified as walking. There were no wing movements except for those associated with occasional grooming bouts.

## Hemilineage 7B

Lineage 7 has an anterior cell cluster located in ventrolateral region of neuromeres T1 through A1 and consists of the surviving 7B hemilineage (*Truman et al., 2010*). The 7B neurite bundle crosses the midline in the iA commissure and then turns anteriorly. The basic morphology is much the same in the adult (*Figure 5H*), as described in *Brown and Truman (2009)*. The neurite projects dorsally, forming a bushy ipsilateral (proximal) arbor in the tectulum, then crosses the midline to form an ascending tract that extends through the neck connective (*Figure 5H*). The arbor in the T2 hemineuromere includes prominent projections into the dorsolateral region of the T2 leg neuropil (*Figure 5I*). The T1 and T3 versions also send a branch into their respective leg neuropils (*Figure 5*, *Figure 5—figure supplement 1*), but these branches are not as robust as the T2 version.

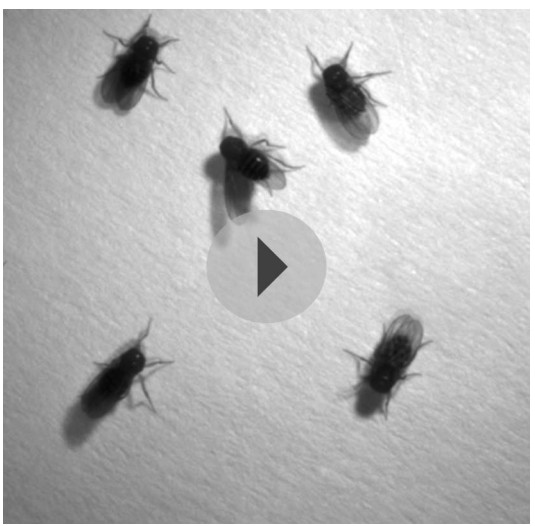

**Video 5.** Behavioral effects of exciting the neurons in hemilineage 5B. This video related to *Figure 4*. Decapitated flies expressing TRPA1 under control of R86D02 are subjected to a 24° to 37°C heat ramp and recorded at 60 fps. Full genotype in *Supplementary file 1*.

The gradual activation of the 7B interneurons by the heat ramp resulted in little walking behavior. The decapitated flies continued spontaneous grooming with occasional wing flicking behavior until they abruptly launch themselves into the air (*Video 8*). If they landed back on the hotplate, they then repeated the behavior. High speed video analysis (*Figure 5J*, *Video 9*) shows that their behavior corresponds to the normal takeoff sequence described by *Card and Dickinson (2008)*: first the wings are raised, the mesothoracic legs then extend in a jump, and finally the wings depress and the fly begins to flap.

## Hemilineage 8A and 8B

In the larva, the lineage 8 neurons are in a ventrolateral cell cluster in the anterior part of each thoracic segment. The 8A neurons project to the dorsal region of the ipsilateral leg neuropil while the 8B neurons extend across the iA commissure, just anterior to the 7B bundle. In the adult, the 8A somata are found in approximately the same location; their bundled neurites make a pronounced lateral bend after they enter the neuropil and form prominent arbors that ramify through the lateral portion of the ipsilateral leg neuropil (*Figure 5K,L*).

The 8B cluster in the adult contains multi-part, midline-crossing, intersegmental arbors (*Figure 5N*). Proximal arbors include an intrasegmental lateral arbor and an ascending medial arbor in the dorsal third of the ipsilateral neuropil. The hemilineage also makes symmetric ascending projections in the upper part of the dorsal neuropil. The T2 and T3 hemilineages send projections to T1, while the T1 hemilineage sends projections up the neck connective. A prominent component of the T3 lineage are the contralateral haltere interneurons (cHINs; *Strausfeld and Seyan, 1985*), with their distinctive 'bowtie' arbor (*Figure 5N,O*). These neurons receive sensory inputs from the haltere nerve, on their lower, intrasegmental ipsilateral arbor, and then make an ascending contralateral projection. The homologous lower arbor is found in T2 where the wing nerve inserts, and may receive wing sensory inputs. A similar, but reduced, arbor is also found in T1.

Our line for the 8B interneurons had some expression from other thoracic hemilineages and was not used for behavioral studies. Activation of the 8A interneurons using line R69H11-GAL4 had minimal effects on the behavior of the decapitated flies (*Figure 5M*, *Video 10*). As the temperature ramped up, we saw little effect on spontaneous behavior. The flies continued to show bouts of grooming; they also showed fidgety repositioning of their legs, but no walking or wing movements.

## Hemilineage 9A

In the larva, the lineage 9 cluster is the dorsal-most cluster in the anterior half of the neuromere, and the surviving 9A neurons project to the medial region of the ipsilateral leg neuropil (*Truman et al., 2010*). In the adult, the 9A somata are found in the middle third of the VNS. They are local leg interneurons that arborize in the ventral leg neuropil (*Figure 5P,Q*), as described in *Brown and Truman (2009)*. These neurons overlap with afferents from leg chordotonal organs (*Harris, 2012*).

Activation of the 9A interneurons resulted in a subtle change in posture in the decapitated flies as their legs became gradually more splayed out during the course of the heat ramp (*Figure 5R*, *Video 11*). There was very little locomotion and bouts of spontaneous grooming were occasionally seen.

## Hemilineage 10B

In the larva, only hemilineage 10B survives (*Truman et al., 2010*). The 10B cell body clusters are just lateral to those of lineage 2 and their neurite bundle projects across the iA commissure. The

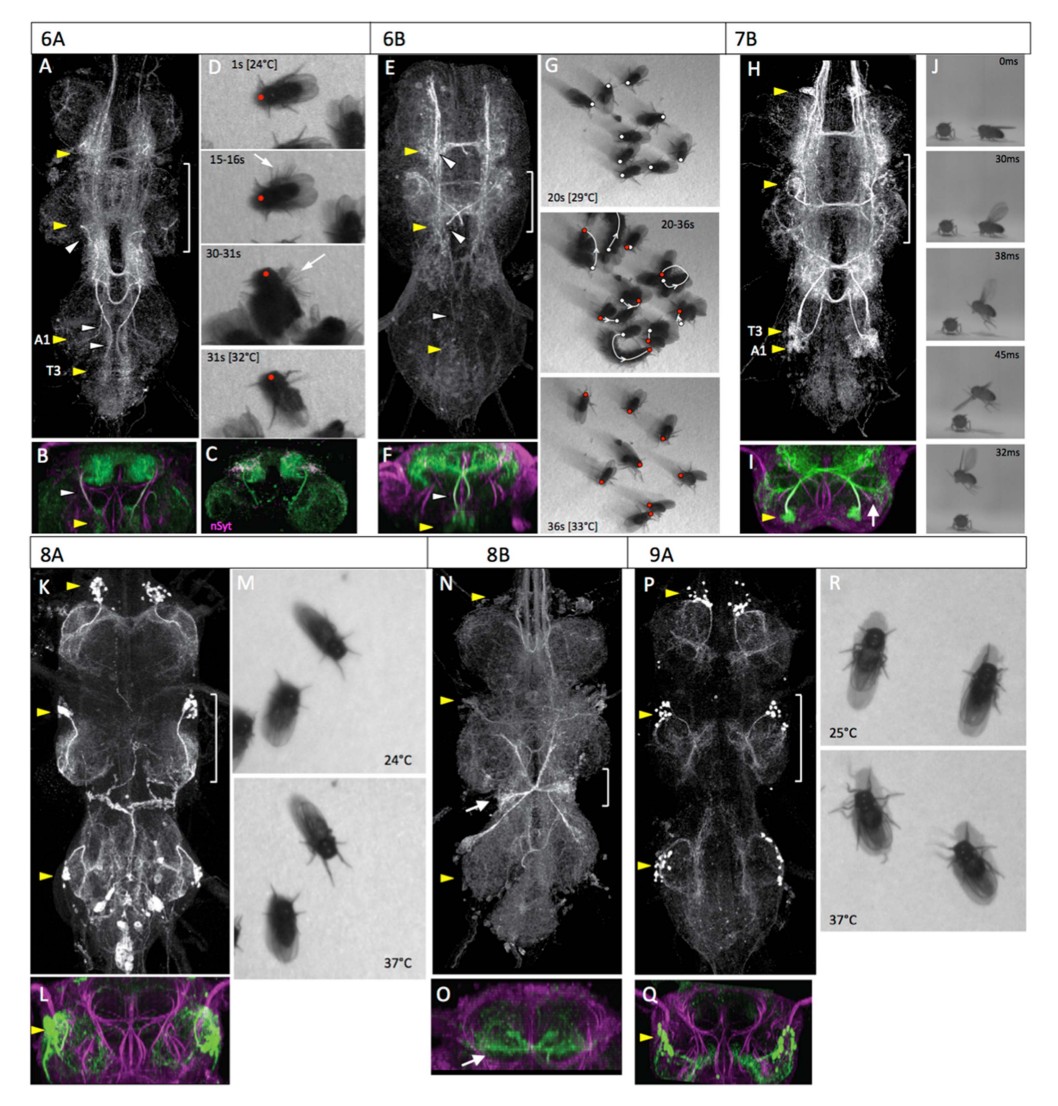

**Figure 5**. The anatomy and behavioral consequences of stimulating hemilineages 6A through 9A. Organization of panels and general symbols are as in *Figure 4*. (**A–D**) Hemilineage 6A. (**A, B**) The 6A neurons are in prominent ventromedial clusters in segments T1 through A1. They ramify through the lateral regions of the tectulum neuropil with a concentration in the dorsal T2 area. (**C**) T2 transverse projection showing that nSyt-GFP (magenta) localizes in the lateral parts of the arbor. (**D**) Video frames of decapitated flies at early (1 s) and intermediate (31 s) times in the heat ramp. Middle frames are multiple exposures taken over 1 s periods showing phasic, repetitive movements of single limbs (15–16 s, arrow), and jerky movements of the entire fly (30–31 s, arrow). (**E–G**) Hemilineage 6B. (**E, F**) The 6B neurons are located in the posterior medial region of each segment and project through a posterior commissure to arborize in the tectulum and dorsal regions of the leg neuropils. (**G**) Video frames showing the position of decapitated flies at an intermediate (20 s) and late (36 s) portion of the heat ramp as they are starting to move. The respective dots mark the anterior margin of the thorax at the 2 times. The middle frame shows the position of the fly at the start (white dot) and end (red dot) of the sequence and at 2 s intervals in between; arrow shows direction of movement. (**H–J**) Hemilineage 7B. (**H, I**) The 7B neurons are in prominent ventrolateral clusters in segments T1 through A1. They ramify through the lateral regions of the tectulum and send a prominent projection into the T2 leg neuropil (arrow). (**J**) Frames of a high-speed video of a decapitated fly taking off during heating. It shows the expected sequence of wing elevation (30 ms), jump (38 ms), and flapping (45 ms). (**K–M**) Hemilineage 8A. (**K, L**) The 8A cluster is situated in the anterolateral region of each segment and projects into the lateral leg neuropil. (**M**). Early and late frames during the heat ramp showing only minor positional changes through the period. (**N, O**) Hemilineage 8B. (**N, O**) The 8B clusters are in the anterolateral region of each segment and project through an anterior commissure to spread widely through the thoracic neuromeres. A prominent subset of the t3 cells form

*Figure 5. continued on next page*

*Figure 5. Continued*

a bowtie-shaped structure (arrow) that receives input from haltere afferents. Transverse section (**O**) is of this input area. (**P–R**) Hemilineage 9A. (**P, Q**) The 9A clusters assume an anterolateral position and project into the region of the ventral leg neuropil that receives input from proprioceptors. (**R**). Video frames from early and late stages of the heat ramp. The decapitated flies responded by splaying out their legs. Genotypes: for most genotypes see *Supplementary file 1*; for the remainder: (**C**) nSyb-GAL80 (su(Hw)attP8)/LexAop-RFP (su(Hw)attP8); UAS-flp (attP40)/ act>STOP>LexA (attP40); R35A03-GAL4 (attP2)/LexAop-nSyt-GFP (su(Hw)attP1).

The following figure supplement is available for figure 5:

**Figure supplement 1**. Dorsal (**A**) and transverse (**B**) view of the adult VNS showing a MARCM clone for the T3 lineage 7.

---

hemilineage 10B cell bodies remain anteromedial in the adult. Their primary neurites cross the midline in an iA bundle and form a finger-like ventral arbor, which starts medial and projects ventrally almost to the leg nerve insertion, and an intersegmental dorsal arbor running just lateral to the midline (*Figure 6B*). The dorsal arbor continues up the neck connective (*Figure 6A*).

The behavioral responses of decapitated flies to the activation of the 10B interneurons were dominated by leg movements (*Figure 6C*; *Video 12*). These movements were somewhat erratic, often causing the flies to make pivoting movements around a leg. When there was net movement, it was usually backwards. Wing flicking and wing buzzing occasionally occurred during walking. Wings were typically held back, rather than in the flight position, during these movements. Rarely a fly would go airborne during one of these episodes.

## Hemilineages 11A and 11B

We were unable to construct lines to target hemilineages 11A and 11B independently, but R26B05-GAL4 targets the whole lineage. Lineage 11A is found in T1 and T2, whereas lineage 11B is found only in T2. Based on this, we attributed the parts of the T2 arbor to 11A vs 11B, based on 11A projections in T1. In the adult, the lineage 11 cell bodies are in the dorsoposterior margin of T1 and T2. The lineage 11 neurons show a complex projection that is predominantly in T2 and largely confined to the tectulum neuropil (*Figure 6D,E*). However, a branch of the 11A arbors from the T1 cells project into the T1 leg neuropil while the corresponding branch from the T2 cells extends into the T3 leg neuropil. There is no equivalent branch for the T2 neuropil.

Similar to the 7B interneurons, the activation of the lineage 11 interneurons using R26B05-GAL4 evoked takeoff behavior. High-speed video analysis showed that the behavioral sequencing differed from normal takeoff in that the decapitated flies commenced wing flapping prior to the jump (*Figure 6F*, *Video 13*). The R26B05 line used to activate the lineage 11 cells occasionally showed expression in a bundle of descending interneurons (*Figure 6—figure supplement 1A*) and we were concerned that their severed axons might be responsible for driving the takeoff behavior in the decapitated flies. Consequently, we used teashirt-GAL80 (tsh-GAL80; *Clyne and Miesenböck, 2008*) to suppress thoracic expression, but leaving expression in the descending interneurons (*Figure 6—figure supplement 1B*). These decapitated flies showed a severe reduction in the number of takeoffs when subjected to the heat ramp (9% of tsh-GAL80 flies [N = 22] vs 100% [N = 13] for R26B05-GAL4 flies lacking the tsh-GAL80).

## Hemilineages 12A and 12B

In the larva, the paired lineage 12 cell body clusters are in the posterior ventromedial region of the thoracic neuromeres. Their neurite bundles project dorsally and then split, with the 12B neurons projecting contralaterally through the iP commissure while the 12A neurons continue into dorsal neuropil. The 12A neurons die in T3. In the adult, the cell body clusters for the 12A and 12B hemilineages remain ventral but move slightly more lateral. Hemilineage 12A arborizes throughout the dorsal tectulum neuropil. The arbors from the T1 and T2 clusters converge on T2 and extensively overlap (*Figure 6G,H*). nSyT::GFP localizes to the medial portions of the dorsal arbors, plus a small region in the lateral part of the lower arbors (*Figure 6I*). The mature 12B neurons have no ipsilateral

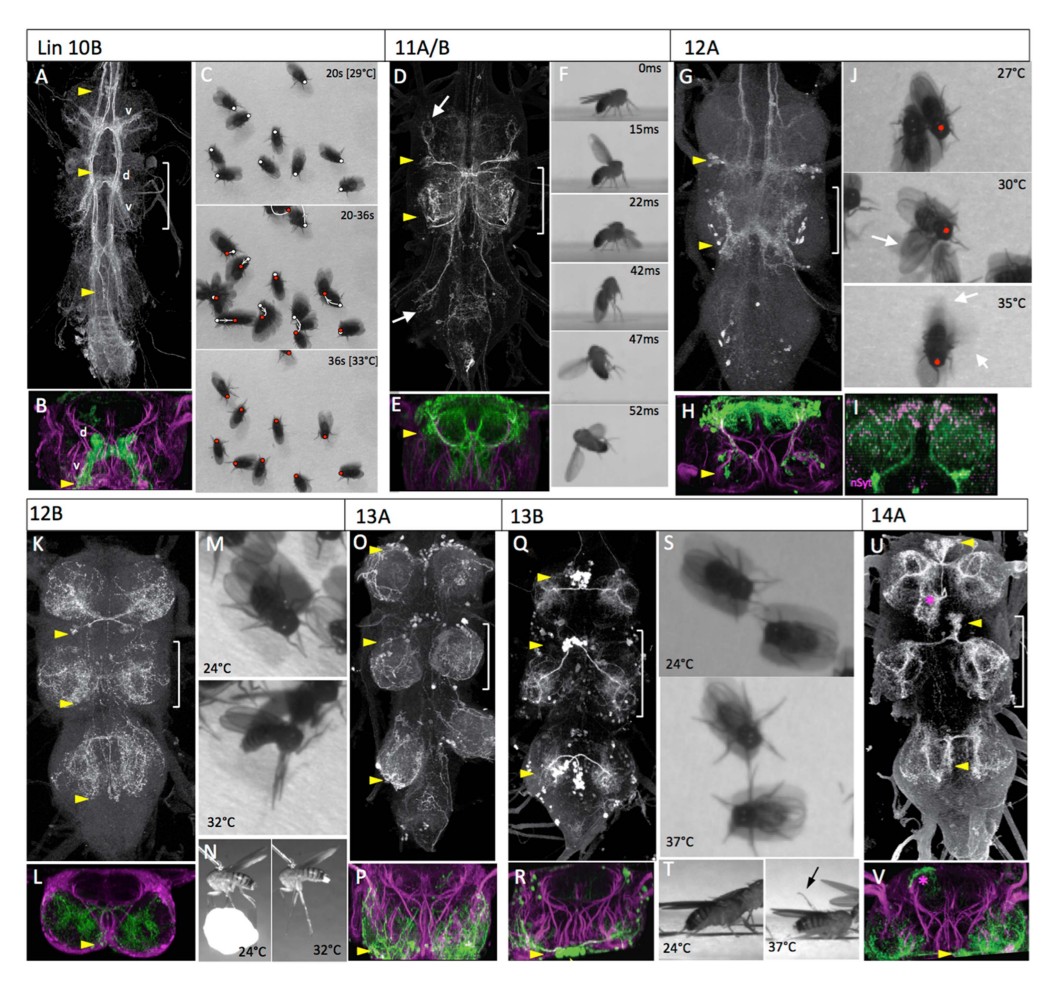

**Figure 6**. The anatomy and behavioral consequences of stimulating hemilineages 10B through 14A. Organization of panels and general symbols are as in *Figure 4*. (**A–C**) Hemilineage 10B. (**A, B**) The 10B clusters are ventromedially located in the anterior part of the segment. They project across an anterior commissure and produce a dorsal (d) arbor that runs longitudinally, and a ventral (v) arbor that extends into medial leg neuropil. (**C**) Video frames showing the position of decapitated flies at intermediate (20 s) and late (36 s) portion of the heat ramp. The respective dots mark the anterior margin of the thorax at the 2 times. The middle frame shows the position of the fly at the start (white dot) and end (red dot) of the sequence and at 2 s intervals in between; arrow shows direction of movement which was generally backward. (**D–F**) Hemilineages 11A and B. (**D, E**) These hemilineages are laterally located only in T1 (11A) and T2 (11A and 11B). They ramify primarily in the T2 tectulum neuropil and have projections into the leg neuropils of T1 and T3 (arrows). (**F**) Frames of a high-speed video of a decapitated fly taking-off during heating. Flapping begins prior to the jump. (**G–J**) Hemilineage 12A. (**G, H**) The ventrolateral 12A clusters are on the posterior border of segments T1 and T2. They project dorsally and arborize through most of the dorsal tectulum. (**I**) T2 transverse projection showing that nSyt-GFP (magenta) localizes to medial regions of the 12A projection. (**J**) Video frames showing progression of behaviors of decapitated flies during the heat ramp. 27°C: flies quiet; 30°C, some walking and showing lateral wing waving (arrow); 35°C: flies showing wing buzzing (blur) although wings (arrows) usually not extended in flight position. (**K–N**) Hemilineage 12B. (**K, L**) The 12B clusters are ventrally located at the posterior border of the neuromere. They project across a posterior commissure and arborize widely through the contralateral leg neuropil. (**M, N**) Video frames of dorsal and lateral views of decapitated flies subjected to a heat ramp. At elevated temperatures the flies often showed tonic extensions of the T2 and T3 legs. (**O, P**) Hemilineage 13A. (**O, P**) The somata of the 13A neurons are spread over the anterior ventrolateral region of the neuromere. Their arbors extend through most of the ventral half of the leg neuropil. (**Q–T**) Hemilineage 13B. (**Q, R**) The 13B clusters are pulled to the anterior midline and their axons cross the midline in a very ventral anterior commissure and the cells branch through the ventrolateral leg neuropil. (**S, T**) Video frames of dorsal and lateral views of decapitated flies early and late in the heat ramp. At high temperatures the legs are extended laterally and often are elevated from the substrate (arrow). (**U, V**) Hemilineage 14A. (**U, V**) The 14A clusters are also pulled to the midline and their axons cross in a ventral commissure. They project through most of the ventral and lateral leg neuropil. This line also expressed in occasional other lineages (*). Genotypes: for most genotypes

*Figure 6. continued on next page*

*Figure 6. Continued*

see *Supplementary file 1*; for the remainder: (**I**) nSyb-GAL80 (su(Hw)attP8)/act>STOP>LexA (attP18); LexAop-RFP (attP40)/UAS-flp (attP40); LexAop-nSyt-GFP (su(Hw)attP1)/R24B02-GAL4 (attP2).

The following figure supplements are available for figure 6:

**Figure supplement 1**. tsh-GAL80 eliminates expression specifically in the VNC.

---

arbor and cross the midline in the pI commissure to arborize in the medial region of the contralateral leg neuropil (*Figure 6K,L*).

Activation of the 12A interneurons produced a sequence of behaviors as the temperature ramped up (*Figure 6J*; *Video 14*). The decapitated flies started with walking behavior that was later joined by wing flicking and by the lateral extension and vibration of a single wing (*Figure 6J*, 30˚C), resembling the courtship singing of the male. Eventually the frequency of wing vibrations became extreme but the wings were usually not extended in the flight position; often one vibrating wing was partially or fully extended to the side while the other was vibrating while over the back (*Figure 6J*, 35˚C). These flies would occasionally go careening into the air, but there did not appear to be an organized jump preceding going airborne.

In contrast to the 12A cells, the activation of the 12B interneurons had little effect on some decapitated flies but evoked a tonic postural change in others. The T2 and T3 legs underwent an extreme extension and the legs froze in this position (*Figure 6M,N*, *Video 15*). These flies either stayed rigidly upright or toppled over onto their side. The movements of the T1 legs were variable.

## Hemilineages 13A and 13B

In the larva, both hemilineages of lineage 13 contribute to the immature leg neuropils; the 13A neurons project to the ipsilateral leg and 13B neurons project across the vA commissure to the contralateral leg neuropil. In the adult, the 13A neurons insert into the neuropil near the entry of the leg nerve. Their arbors extend along the edges of the leg neuropil and ramify through its ventral half (*Figure 6O,P*). The cell bodies of 13B cluster are pulled medially during metamorphosis and sometimes are pulled across the midline (*Figure 6Q*). Their axons cross the midline in the vA commissure and constitute the ventral-most bundle (*Figure 6R*). The 13B neurons branch

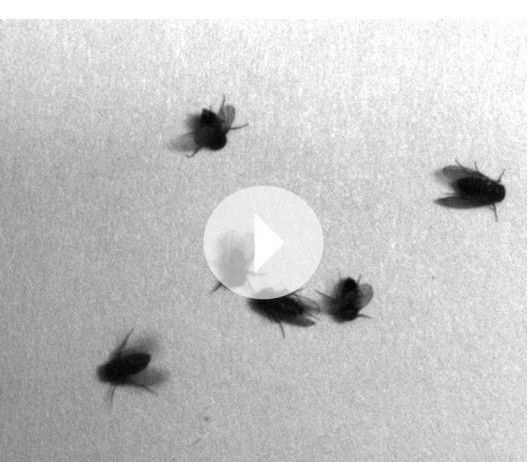

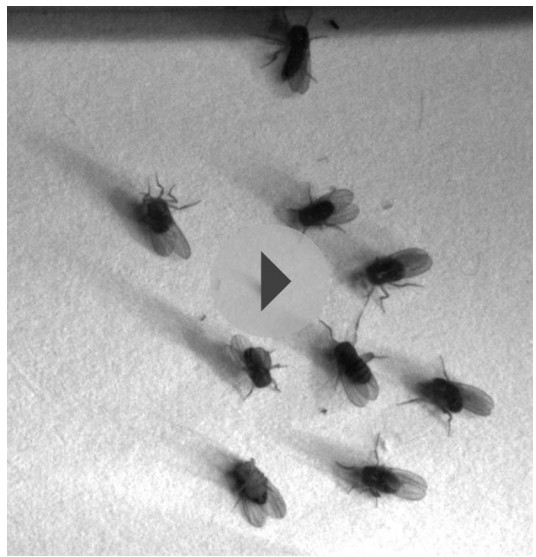

**Video 6.** Behavioral effects of exciting the neurons in hemilineage 6A. This video related to *Figure 5*. Decapitated flies expressing TRPA1 under control of R35A03 are subjected to a 24˚ to 37˚C heat ramp and recorded at 60 fps. Full genotype in *Supplementary file 1*.

**Video 7.** Behavioral effects of exciting the neurons in hemilineage 6B. This video related to *Figure 5*. Decapitated flies expressing TRPA1 under control of R46C11 are subjected to a 24˚ to 37˚C heat ramp and recorded at 60 fps. Full genotype in *Supplementary file 1*.

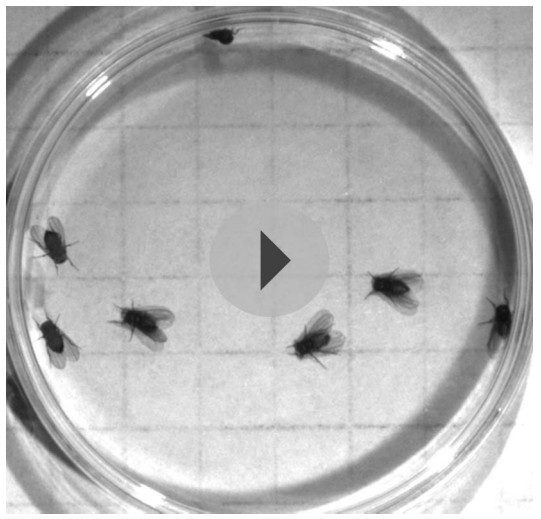

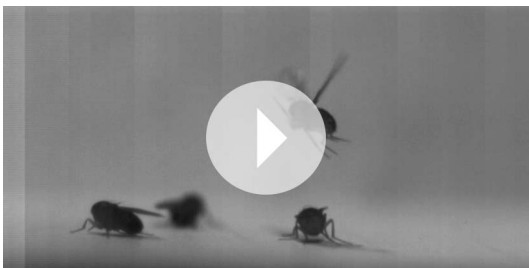

**Video 9.** High-speed video showing the behavioral effect of exciting the neurons in hemilineage 7B. This video related to *Figure 5*. Decapitated flies expressing TRPA1 under control of R65A12 are subjected to a 24° to 32°C heat ramp and recorded at 3000 fps. Full genotype in *Supplementary file 1*.

**Video 8.** Behavioral effects of exciting the neurons in hemilineage 7B. This video related to *Figure 5*. Decapitated flies expressing TRPA1 under control of R65A12 are subjected to a 24° to 32°C heat ramp and recorded at 60 fps. Full genotype in *Supplementary file 1*.

extensively through the ventral region of the leg neuropil.

We did not have a clean enough line to assess the effects of activating the 13A interneurons. Activation of the 13B cells in decapitated flies evoked a tonic postural change. As the temperature increased they extended their legs progressively more to the side so that by the end of the heat ramp they were often resting on their coxae, with their tarsi elevated into the air (*Figure 6S,T*; *Video 16*).

## Hemilineage 14A

In lineage 14, most of the 14B neurons die, so the cluster is comprised almost exclusively of hemilineage 14A neurons (*Truman et al., 2010*). During metamorphosis the cell clusters of the 14A neurons are pulled to the anterior midline. Their axons form an extreme ventral bundle that is just

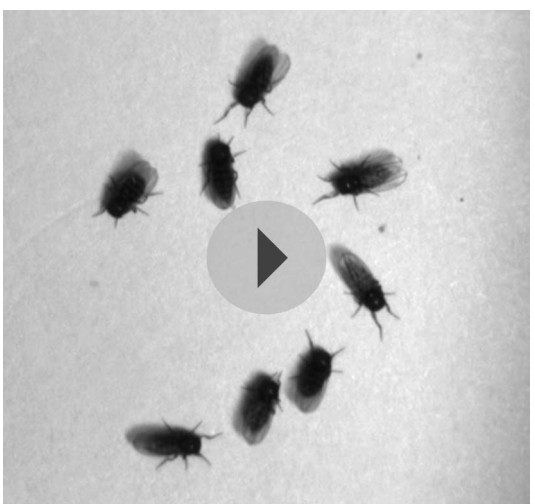

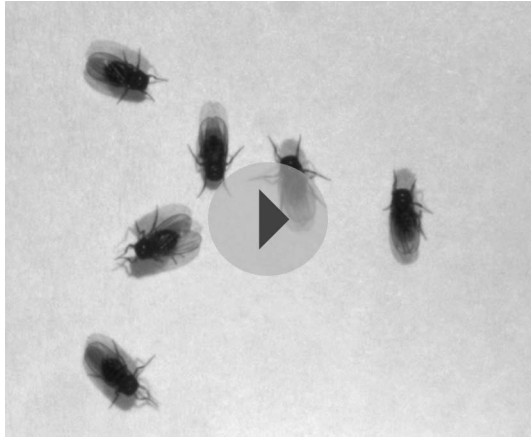

**Video 10.** Behavioral effects of exciting the neurons in hemilineage 8A. This video related to *Figure 5*. Decapitated flies expressing TRPA1 under control of R69H11 are subjected to a 24° to 32°C heat ramp and recorded at 60 fps. Full genotype in *Supplementary file 1*.

**Video 11.** Behavioral effects of exciting the neurons in hemilineage 9A. This video related to *Figure 5*. Decapitated flies expressing TRPA1 under control of R52E12 are subjected to a 24° to 37°C heat ramp and recorded at 60 frames per second (fps). Full genotype in *Supplementary file 1*.

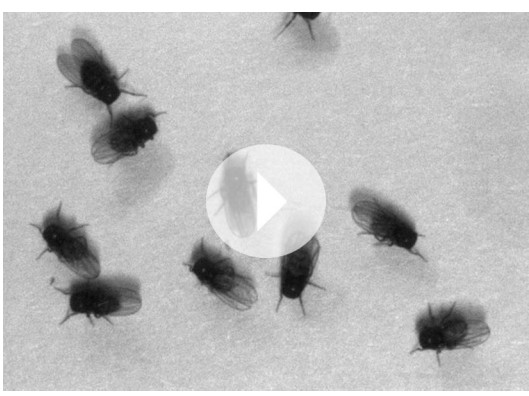

**Video 12.** Behavioral effects of exciting the neurons in hemilineage 10B. This video related to *Figure 6*. Decapitated flies expressing TRPA1 under control of R13B08 are subjected to a 24° to 32°C heat ramp and recorded at 30 fps. Full genotype in *Supplementary file 1*.

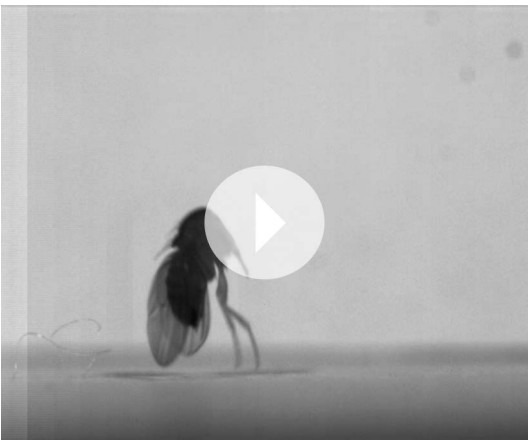

**Video 13.** High-speed video showing the behavioral effect of exciting the neurons in hemilineages 11A and B. This video related to *Figure 6*. Decapitated flies expressing TRPA1 under control of R26B05 are subjected to a 24° to 37°C heat ramp and recorded at 6400 fps. Full genotype in *Supplementary file 1*.

dorsal and anterior to the 13B neurons. The 14A neurons have no ipsilateral arbor and arborize throughout the ventral half of the leg neuropil and also laterally in the region where the motoneurons arborize (*Figure 6U,V*). Our best line for 14A interneurons had major contamination with the 2A hemilineage, so we do not have behavioral data for this hemilineage.

## Hemilineage 15B

This is one of the two major lineages that make leg motoneurons (*Figure 7A,B*). The adult composition of this lineage has been extensively characterized (*Baek and Mann, 2009*; *Brierley et al., 2012*) and they primarily supply muscles of the distal leg segments.

Our best 15B lines had contamination from some of the interneuron hemilineages in the thorax. Consequently, we do not have behavioral observations for this hemilineage.

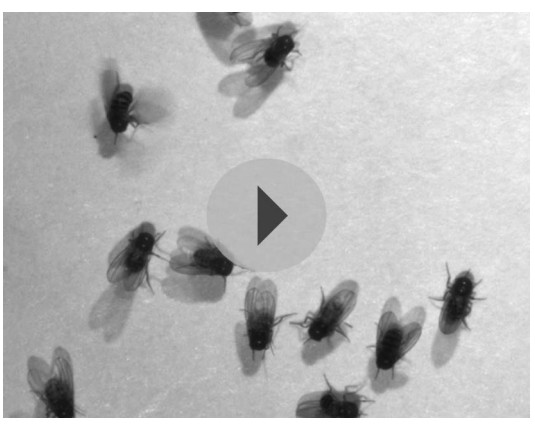

**Video 14.** Behavioral effects of exciting the neurons in hemilineage 12A. This video related to *Figure 6*. De-capitated flies expressing TRPA1 under control of R24B02 are subjected to a 24° to 37°C heat ramp and recorded at 60 fps. Full genotype in *Supplementary file 1*.

## Hemilineage 18B

Lineage 18 is missing from T1 and only the 18B hemilineage persists in T2 and T3 (*Truman et al., 2010*). In the larva, the 18B clusters are situated at the anteriodorsal margin of the segment and they send their axons across the iA commissure and into a longitudinal tract. The adult projection pattern is complex as the cells project through much of the tectulum neuropil (*Figure 7C,D*). The T2 neurons show a concentration of arbor in the dorsal-most regions of the tectulum and also project a ventral arbor into the dorsolateral regions of the leg neuropils, a region occupied by the dendrites of the leg motoneurons. The T3 arbor is similar to that of T2, but reduced, especially with regard to its projection to the leg neuropil.

In response to the activation of the 18B neurons by the heat ramp, the decapitated flies typically initiated walking and that was some-times accompanied by jerky wing movements

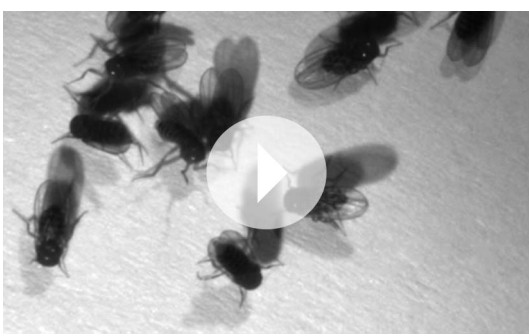

**Video 15.** Behavioral effects of exciting the neurons in hemilineage 12B. This video related to *Figure 6*. Decapitated flies expressing TRPA1 under control of R15D11 are subjected to a 24° to 37°C heat ramp and recorded at 60 fps. Full genotype in *Supplementary file 1*.

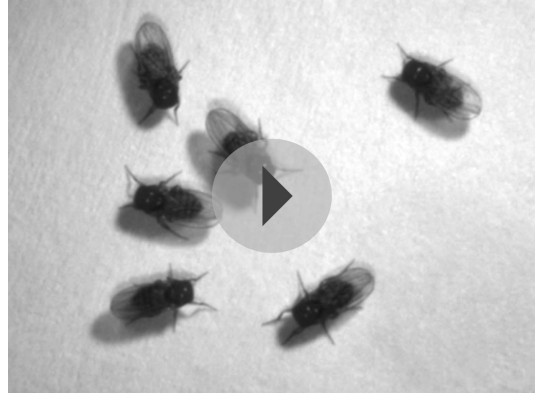

**Video 16.** Behavioral effects of exciting the neurons in hemilineage 13B. This video related to *Figure 6*. Decapitated flies expressing TRPA1 under control of R41G09 are subjected to a 24° to 32°C heat ramp and recorded at 60 fps. Full genotype in *Supplementary file 1*.

with the wings remaining partially folded. After a latent period (varying from milliseconds to seconds), the fly would jump and the wing movements would transition into flapping, but not consistently in that order (*Figure 7E*; *Video 17*).

## Hemilineages 19A and 19B

In the larva, the lineage 19 cluster is situated dorsolaterally at the posterior border of each thoracic neuromere. The 19A neurons descend into the ipsilateral leg neuropil, while the 19B neurons project across the iP commissure and turn anteriorly. The adult morphology is similar. The 19A and 19B cell clusters remain in the posteriodorsal region of each segment. The 19A neurons project into the ipsilateral leg neuropil, where they form a major projection extending through lateral and ventral leg neuropil and a medial projection that extends to the midline. The medial projections in T1 and T2 converge on the midline in T2 (*Figure 7F,G*). nSyt::GFP localizes to the most medial (distal) part of the medial projection (*Figure 7H*). The hemilineage 19B cluster in T1 is greatly reduced in the larva, and after metamorphosis the 19B cluster in T3 is also very small, presumably through cell death during metamorphosis. The adult T2 neurons make a robust projection across the iP commissure and arborize dorsally in the tectulum neuropil. The 19B neurons have both medial and lateral arbors that project anteriorly (*Figure 7J,K*; *Figure 7—figure supplement 1*).

Our best 19B line had massive contamination from the haltere afferents and so we have no behavioral data for this lineage. During activation of the 19A interneurons the decapitated flies stayed in place and showed no movements of their T1 or T3 legs, but their T2 legs show incessant waving movements that continue for the duration of the thermal activation (*Figure 7I*; *Video 18*).

## Hemilineages 20A and 22A

In the larva, 20A and 22A are adjoining lineages in the posterior half of the neuromere. Their first few B progeny become motoneurons but the remainder die (*Truman et al. 2010*). The A progeny of both NBs send very short projections to the ventral leg neuropil, and it is difficult to tell them apart. We were unable to recover lines that distinguished them, and have treated them as a single entity in the adult. The cell body clusters for 20A/22A are found at the posteriolateral border of each thoracic segment (*Figure 7L*). The clusters project anteriorly into the mid region of the leg neuropil and then make a robust lateral branch and a thinner medial branch that extends into dorsal leg neuropil (*Figure 7M*).

Activation of the 20A/22A interneurons evoked a simple change in posture in the decapitated flies. As the temperature increased, the flies extended their legs until they had assumed a splayed out posture (*Figure 7N*; *Video 19*). Occasional grooming bouts occurred during and after the change in posture.

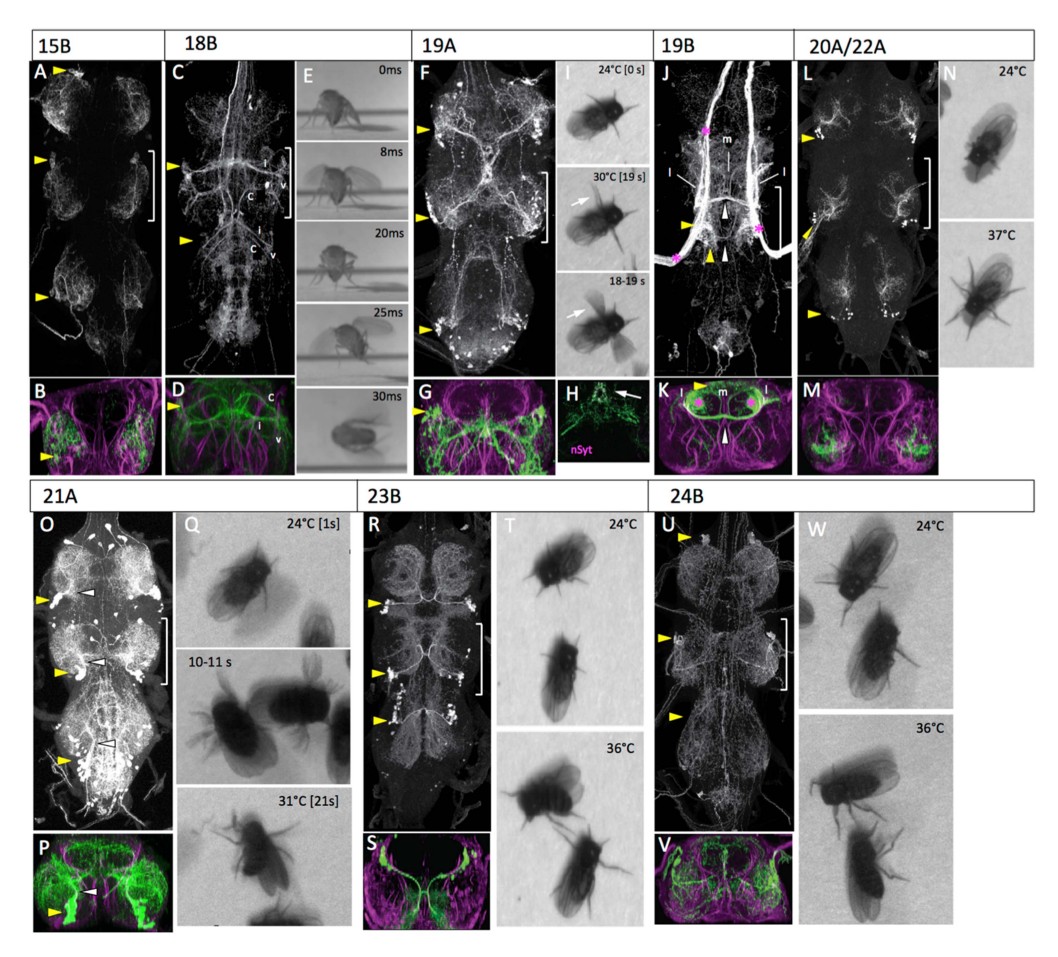

**Figure 7**. The anatomy and behavioral consequences of stimulating hemilineages 15B through 24B. Organization of panels and general symbols are as in *Figure 4*. (**A–C**) Hemilineage 15B. (**A, B**) The larger of the two motor lineages. Cell bodies are located in the anterolateral region of the segment and the neurons innervate primarily muscles to distal leg segments. (**C–E**) Hemilineage 18B. (**C, D**) The 18B clusters are located in the dorsoanterior regions of T2 and T3. Their axons cross in an anterior commissure and have discrete contralateral (c), intermediate (i) and ventral (v) projections. The ventral projections extend into the leg neuropil. (**E**) Frames of a high-speed video of a decapitated fly taking-off during heating. Flapping often began prior to wing raising causing the wings to bend (8ms, 25ms). (**F–I**) Hemilineage 19A. (**F, G**) The 19A clusters are posterior dorsolateral in the segment and project arbor into the ventral leg neuropil and also to a convergence point at the midline just below the tectulum. This convergence point shows nSyt-GFP (magenta) localization (**H**, arrow). (**I**) Video frames from early and midway through the heat ramp. With increased temperature the flies extend their T2 legs and begin to wave them. The frames for a one second period (18–19 s) are superimposed to illustrate movements of the T2 legs (arrow) and stability of the others. (**J, K**) Hemilineage 19B. (**K, L**) The major representation of the 19A cells are in the T2 cluster located posterior dorsolateral region of the segment, with a smaller cluster in T3. The axons cross in a posterior commissure and show medial (m) and lateral (l) anterior projections that are confined to the tectulum. The line had massive contamination from haltere afferents (*). (**L–N**) Hemilineages 20A, 22A. (**L, M**) The cell bodies for the 20A, 22A clusters are in the posterior ventrolateral region of the segment and the neurons innervate the middle third of the leg neuropil. (**N**) Video frames of decapitated flies showing that they splay out their legs as the temperature rises. (**O–Q**) Hemilineage 21A. (**O, P**) The 21A clusters are in the posterior ventrolateral region of the segment. They project dorsomedially and then arborize over most of the dorsal two-thirds of the leg neuropil. (**Q**) Video frames of decapitated flies during the heat ramp. Eventually the flies become immobile with their legs frozen at unusual angles (21 s). The frames for an intermediate, one second period (10–11 s) are superimposed to show the transient hyperkinetic leg movements. (**R–T**) Hemilineage 23B. (**R, S**) The 23B clusters are in the posterior dorsolateral region of the segment. The neurons produce an extensive ipsilateral arbor and then cross the posterior commissure for their output arbor. (**T**) Stimulation of the 23B neurons produce uncoordinated leg movements and the flies basically

*Figure 7. continued on next page*

*Figure 7. Continued*

stay in place (36°C). (**U–W**) Hemilineage 24. (**U, V**) The smaller of the two motor lineages. Cell bodies are located in the anterodorsolateral region of the segment and the neurons innervate primarily proximal leg muscles. (**W**) Video frames of decapitated flies subjected to a heat ramp. At elevated temperatures the flies typically showed repetitive leg movements. Genotypes. For most genotypes see *Supplementary file 1*; for the remainder: (**I**) UAS-nSyt-GFP (attp18)/w; UAS-RFP(attP40)/+: R32E04_GAL4 (attP2).

The following figure supplement is available for figure 7:

**Figure supplement 1**. Dorsal (**A**) and transverse (**B**) view of the T2 region of the adult VNS showing a lineage 19 MARCM clone.

## Hemilineage 21A

In the larva, lineage 21 is situated next to 20 and 22 but the 21A neurons project into the medial region of the immature leg neuropil, rather than staying lateral. In the adult, the cell body cluster of the 21A neurons remains close to the 20A/22A cells. They project dorsally into the middle of the ipsilateral leg neuropil and send branches through most of the leg neuropil (*Figure 7O,P*).

Activation of the 21A interneurons using the R51H05-GAL4 line induced uncoordinated leg movements that lacked both intralimb and interlimb coordination. The leg movements were incessant during the stimulation but, typically, the flies did not move from their place (*Figure 7Q*; *Video 20*).

## Hemilineage 23B

In the larva, only the hemilineage 23B siblings survive (*Truman et al., 2010*). The hemilineage 23B cell bodies are located ventrolateral in the larva but are displaced dorsally during metamorphosis. The adult 23B neurons project ventrally in parallel with the axons of the 19B neurons and cross the iP commissure. The 23B cluster sends off a robust arbor just prior to crossing the midline (*Figure 7R*). This ipsilateral arbor extends through the ventralmost portions of the leg neuropil, where it appears to overlap with multiple classes of leg sensory neurons (*Figure 7S*). The contralateral arbor extends to dorsal leg neuropil in adjacent segments.

Activation of the 23B interneurons using line R77C10-GAL4 caused a progressive higher

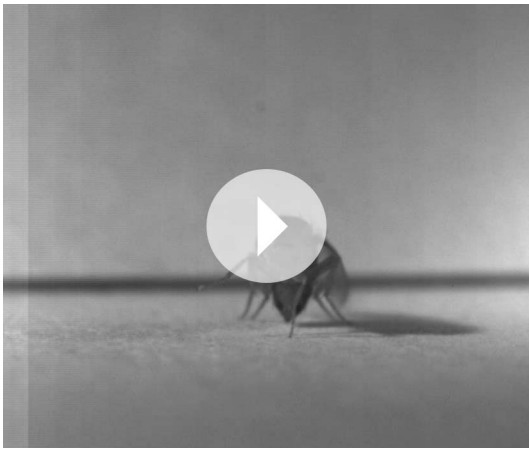 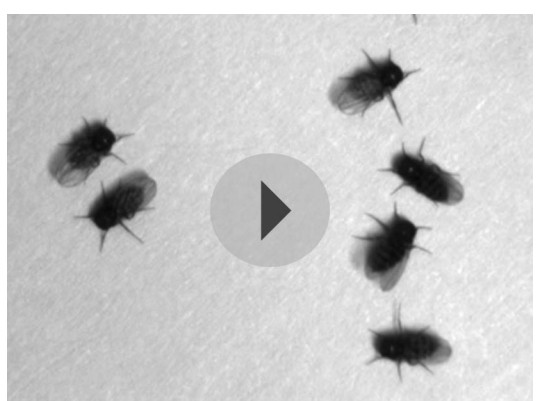

**Video 17.** Behavioral effects of exciting the neurons in hemilineage 18B. This video related to *Figure 7*. Decapitated flies expressing TRPA1 under control of R27A09 are subjected to a 24° to 37°C heat ramp and recorded at 60 fps. Full genotype in *Supplementary file 1*.

**Video 18.** Behavioral effects of exciting the neurons in hemilineage 19A. This video related to *Figure 7*. Decapitated flies expressing TRPA1 under control of R32E04 are subjected to a 24° to 32°C heat ramp and recorded at 60 fps. Full genotype in *Supplementary file 1*.

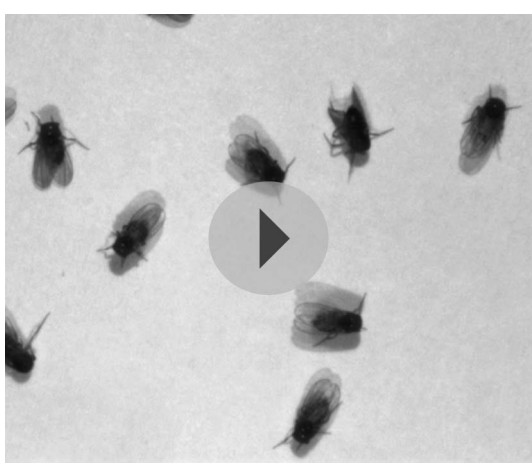

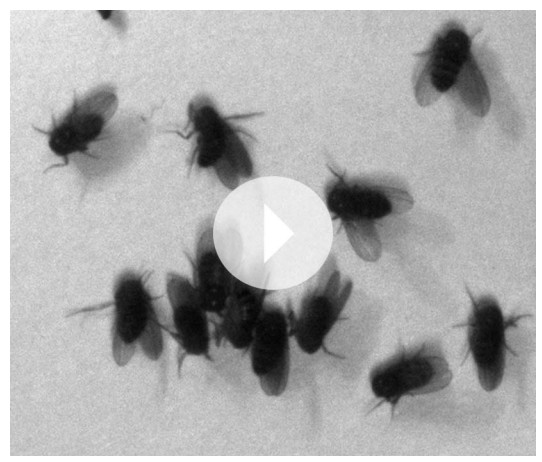

**Video 19.** Behavioral effects of exciting the neurons in hemilineages 20A and 22A. This video related to *Figure 7*. Decapitated flies expressing TRPA1 under control of R24G06 are subjected to a 24˚ to 32˚C heat ramp and recorded at 60 fps. Full genotype in *Supplementary file 1*.

**Video 20.** Behavioral effects of exciting the neurons in hemilineage 21A. This video related to *Figure 7*. Decapitated flies expressing TRPA1 under control of R51H05 are subjected to a 24˚ to 32˚C heat ramp and recorded at 30 fps. Full genotype in *Supplementary file 1*.

frequency of intersegmental limb movements but the intralimb coordination of limb joints appeared to be quite poor (*Figure 7T*; *Video 21*).

## Hemilineage 24B

Hemilineage 24A (*Brown and Truman, 2009*; lineage B of *Baek and Mann, 2009*) is also a motor lineage in which all of the B daughters are leg motoneurons. It is a smaller motor lineage than hemilineage 15B and its cells innervate more proximal leg segments (*Baek and Mann, 2009*). The adult morphology of the 24B motoneurons have been described in detail (*Baek and Mann, 2009*; *Brierley et al., 2012*), and their dendrites occupy the intermediate region of the leg neuropil (*Figure 7U,V*).

The 24B motoneurons were activated using the R15A03-GAL4 line. Stimulation of the 24B motoneurons produced repetitive movements of the legs although these typically resulted in no net movement of the fly (*Figure 7W*; *Video 22*). By the end of the ramp the legs were often held at awkward angles. Rare wing movements occurred but these were usually associated with grooming attempts.

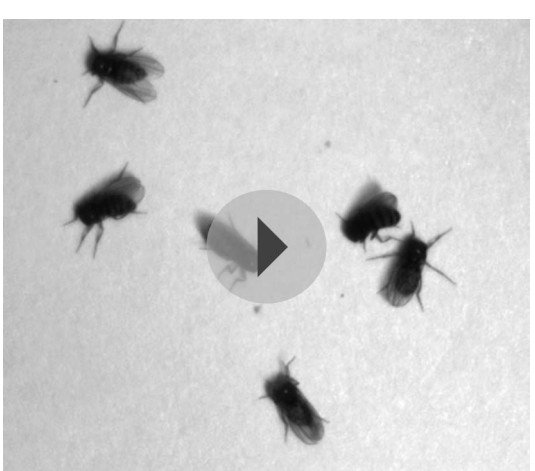

**Video 21.** Behavioral effects of exciting the neurons in hemilineage 23B. This video related to *Figure 7*. Decapitated flies expressing TRPA1 under control of R77C10 are subjected to a 24˚ to 32˚C heat ramp and recorded at 60 fps. Full genotype in *Supplementary file 1*.

## Using hemilineage lines to target behaviorally-relevant groups of neurons

The toolset described here can be used to dissect broad gene expression patterns in the VNS into subsets of cells that come from a common NB, and/or to assign the NB of origin to a neural class of interest. As proof of principle, we demonstrated that a previously identified subset of the *fruitless (fru)*-expressing neurons involved in courtship song are all produced in hemilineage 12A and cleanly isolated this *fru+* cluster.

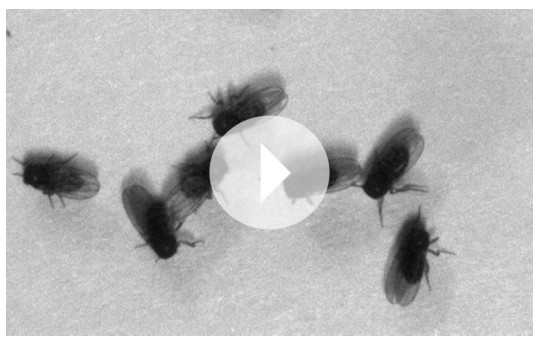

**Video 22.** Behavioral effects of exciting the motor-neurons in hemilineage 24B. This video related to *Figure 7*. Decapitated flies expressing TRPA1 under control of R22G11 are subjected to a 24° to 32°C heat ramp and recorded at 30 fps. Full genotype in *Supplementary file 1*.

The *fru*-expressing neurons are necessary and sufficient to drive most aspects of male courtship behavior, and their anatomy, genetics, and function have been dissected in detail (e.g., *Manoli and Baker, 2004*; *Yu et al., 2010b*; *Meissner et al., 2011*). The *fru* expression pattern can be divided into ~100 groups of cells, including ~40 in the VNS (Yu et al., 2010). *von Philipsborn et al. (2011)* subsequently identified lines targeting those groups using a screen of ~1000 GAL4 lines intersected with *fru*-flippase. The largest cell cluster in the VNS, called vPR6, projects to a structure called the thoracic triangle and appears to be involved in shaping the courtship song (*von Philipsborn et al., 2011*). The latter GAL4 screen recovered 5 lines that each captured 2–5 of these cells in the male and none in the female. No single line hit all of the cells in the vPR6 cluster since *Yu et al. (2010b)* estimated that it should contain approximately 6–10 cells in the male and 4–6 cells in the female.

The location and projection path of the vPR6 cells suggested that they were part of hemilineage 12A. We used the hemilineage 12A combination (including the 12A driver R24B02-GAL4, nSyb-GAL80, UAS-flippase, and pJFRC40-13XLexAop2-FRT>-STOP-FRT>-myr::GFP [*Pfeiffer et al., 2010*; *Nern et al., 2011*]) to restrict 13XLexAop2>myr::GFP expression in the VNS to the neurons of hemilineage 12A, then used *fru*P1.LexA (*Mellert et al., 2010*) to drive GFP expression in the subset of 12A neurons that were part of the *fru* pattern. This technique reliably captured a group of approximately 12 cells in males and 3–5 cells in females. The collected arbor shape of the fru+ hemilineage 12A neurons in both sexes matched the digitally masked arbor assigned to the vPR6 cluster (*Figure 8*) (*Yu et al., 2010b*). Thus, it is possible to identify the developmental origin of secondary neurons of interest and genetically isolate those neurons using the toolkit presented here.

## Discussion

### Hemilineage groups and neuronal anatomy

Our study focused on the anatomy and function of the hemilineage clusters of the thoracic CNS. We developed new genetic tools that allowed us to target most of the 31 interneuron clusters so that we could examine their form and function after metamorphosis. We find that the initial projections that these hemilineage groups make in the larva (*Truman et al., 2004*) prefigure their target regions in the adult CNS. Two motor hemilineages (15B and 24B) and fifteen interneuron hemilineages form the immature 'leg neuropil' of the larva (*Truman et al., 2004*) and all of these make neurons exclusively for the adult leg neuropil. The only mature leg hemilineage that resides outside of the larval leg neuropil is hemilineage 23B whose mature neurons have dendrites in the most ventral regions of the leg neuropil where they overlap the leg afferents. They may be excluded from the immature leg neuropil because their inputs do not grow into the CNS until metamorphosis is well underway.

The neurons in many of the leg hemilineages constitute a relatively homogeneous group with arbors in restricted regions of the leg neuropil. Presumably each of these is dedicated to a particular step in sensory-motor processing for the leg. On the sensory side, the dendrites of the above-mentioned 23B neurons and the neurons of the 9A group overlap the terminals of exteroceptors and proprioceptors, respectively (*Harris, 2012*) and are likely involved with dealing with primary sensory input. More complex computations are likely carried out by the 14A interneurons. Their NB is the persisting NB4-1 NB from the embryo (*Birkholz et al., 2015*), and the homologous NB4-1 in grasshoppers produces the cluster of midline spiking interneurons (*Shepherd and Laurent, 1992*) that conveys information from the somatosensory map to the myotopic motor map (*Burrows and Newland, 1993*; *Burrows, 1996*). The neurons generated by NB 4-1 in flies are anatomically similar to the grasshopper neurons and we expect that the fly neurons are likewise involved in maintaining spatial information during the transformation from the sensory (*Murphey et al., 1989*) to motor maps

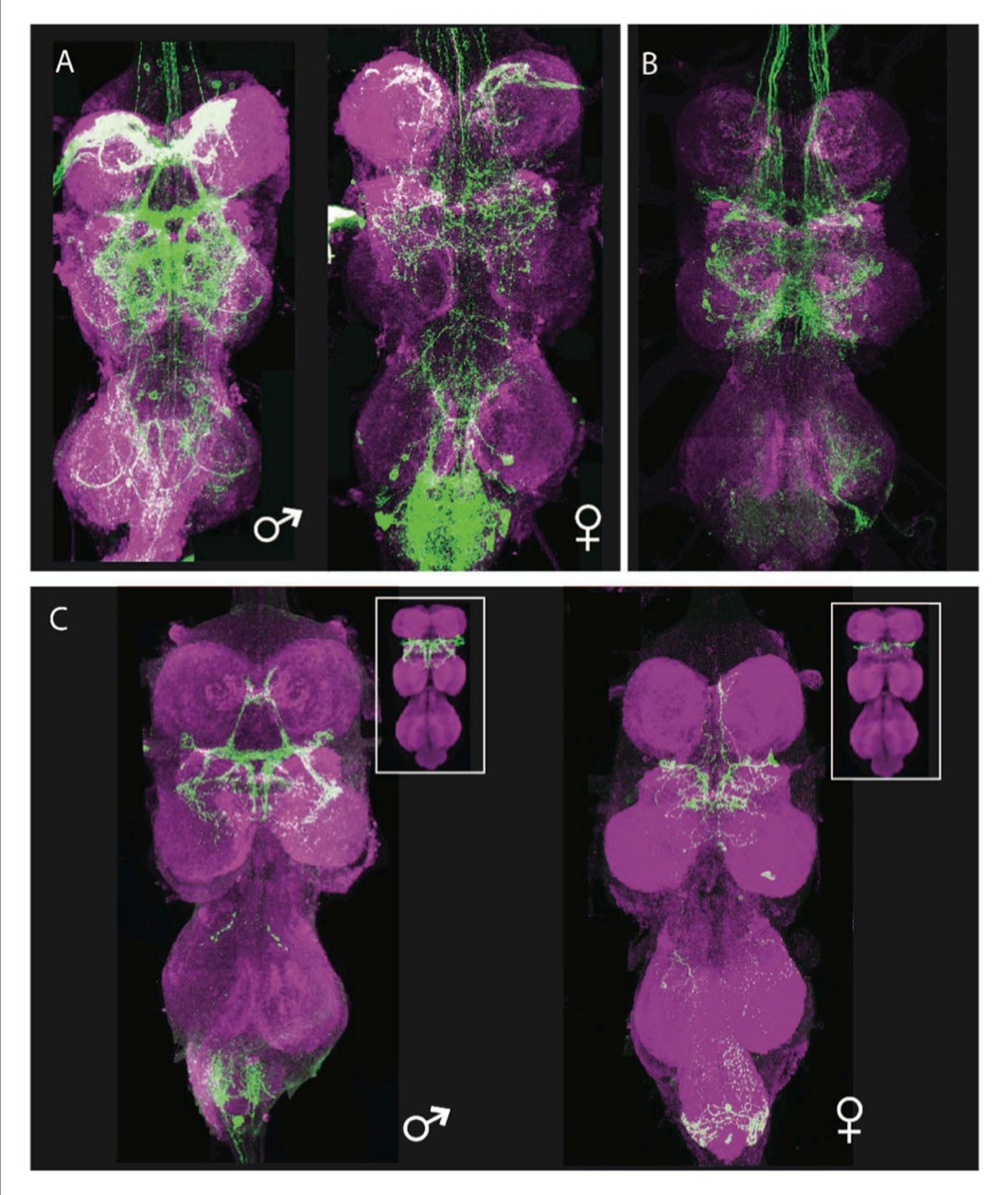

**Figure 8**. Generation of a vPR6-specific line by intersecting hemilineage 12A and fru-LexA. (**A**, **B**) Parental expression patterns. (**A**) The fru-LexA expression pattern. The female thoracic pattern is a subset of the male pattern, and many of the neurons have dimorphic arbors. (**B**) Hemilineage 12A. Genotype: nSyb-GAL80 (su(Hw)attP8)/+; UAS-flp (attP40)/+; R24B02-GAL4/nSyb-LexA, LexAop>STOP>GFP. (**C**) The intersection of A and B, isolating the vPR6 neurons. Genotype: nSyb-GAL80(su(Hw)attP8); UASflp(attP40)/LexAop>STOP>GFP(attP40); R24B02-GAL4(attP2)/ fru-LexA. Insets: the arbors and numbers of cells match digital representations of the complete vPR6 pattern in males (left) and females (right), adapted from *Yu et al. 2010a*.

(*Brierley et al., 2012*) in the leg. Outside of these spatially restricted leg hemilineages, other leg hemilineage clusters have a rather diffuse projection through the leg neuropil, as in the case of 3A and 4B, or they have arbors that also extend beyond their respective leg neuropil, such seen for the 1A and 19A neurons.

The tectulum neuropil is built from the dorsal-projecting lineages in the larva. They arise as segmentally discrete units but as metamorphosis approaches the segmental hemilineages begin to

converge on the T2 region. The hemilineage groups that project to this region are either intrasegmental, joining left and right neuropils, or intersegmental. For some of the dorsal-projecting hemilineages the members in the cluster appear relatively homogeneous (e.g., 6A and 6B), as seen for most of the leg hemilineages. In other instances, though, such as the 12A and 8B groups, the hemilineage cluster is more complex and contains obvious subclasses of interneurons.

The thoracic hemilineages, then, vary in the heterogeneity of neurons that they contain. We speculate that the more homogenous hemilineages are collections of parallel components, which provide parallel but overlapping channels in the circuits that control sensory to motor transformations. The more heterogeneous hemilineages likely have more diversified integrative functions. The recent characterization of lineages in the fly brain (*Ito et al., 2013*; *Yu et al., 2013*) shows many brain lineages that contain a greater diversity of cell types than we see for the VNS hemilineages. Nevertheless, some of the brain lineages also appear to be simple, rather homogeneous collections of neurons and we would expect these to also represent parallel processing units. This idea is supported by a projection neuron hemilineage from the lateral antennal lobe (lAL) NB. Overall, this hemilineage contains five classes of projection neurons but they are all involved in receiving primary sensory input (olfactory, gustatory, or primary antennal mechanosensory) and relaying it to higher centers within or around the mushroom bodies (*Lin et al., 2012*).

## Relationship of behavior to anatomy

We stimulated the neurons in the hemilineages via TRPA1 activation to assess the functional roles of these interneuron groups. Would the stimulation of neurons in different hemilineage groups result in distinctive behavioral responses, and was there any evidence for a functional hierarchy amongst the various groups? As summarized in *Supplementary file 1*, most of our lines had some degree of contamination with expression in neurons that were outside of our target hemilineage. We did not use lines that showed expression in sensory neurons (such as hemilineages 3A and 19B), or that had expression in other hemilineage clusters (e.g., the hemilineage 14A line). We used lines that had expression in a few abdominal neurons per neuromere or weak expression in a few scattered thoracic neurons per neuromere. In all cases, the predominant expression in the line was from the target hemilineage. The problem of extraneous brain contamination was removed by decapitation, although we had concerns about severed descending axons that might still be activated by the heat ramp. Where this was a potential issue (i.e., presence of descending axons in the line) we circumvented the problem by ageing the animals for 24 hr after the decapitation to allow the severed axons to degenerate prior to behavior testing, or by blocking thoracic expression using tsh-GAL80 (*Figure 7*; *Figure 6—figure supplement 1*) and showing that the behavioral response disappeared. We collected behavioral data for 22 of the 33 hemilineage groups.

Another issue was how much of a given hemilineage is captured in the various driver lines? We do not have quantitative data on this question, but comparison of the driver lines expression with MARCM clones for the various lineages (*Brown and Truman, 2009*; *Harris, 2012*; *Figure 4—figure supplement 1*, *Figure 5—figure supplement 1*, *Figure 7—figure supplement 1*) showed that all of the neuropil features evident in the clones were also found in the lines. Therefore, the cellular diversity in the various hemilineage groups is captured in the driver lines that we used.

Our behavior testing system was quite artificial since it utilized decapitated flies. Despite lacking their heads, these flies showed a complex array of behaviors during the tests. They provide insights in how the circuitry of the thorax can operate when freed from the descending influences of the brain and subesophageal ganglia.

Using TRPA1 expression and a heat ramp to activate the hemilineage groups resulted in both tonic and sequential types of behavioral responses. Tonic responses were the more common and are defined as a persistent behavioral response that began at some point during the heat ramp and were then maintained for the remainder of the stimulus. This type of response was seen for changes in posture (such as seen for hemilineages 9A or 13B) for sustained walking (as for the 1A or the 10B neurons), and for sustained wing buzzing (the 2A interneurons). The less common pattern was for the decapitated flies to display an ordered sequence of behavioral responses that changed as the temperature increased. This was very evident for the stimulation of the 12A neurons that began with locomotion that was later joined by lateral wing waving and then finally by the onset of sustained wing buzzing. This behavioral sequencing might result from a progressive increase in neuronal firing rate of the cells in the group during the heat ramp, or by the successive recruitment of group members with higher firing thresholds. We have no data to favor either option at present.

A number of observations come from the analysis of the responses to the activation of the different hemilineage groups (*Figure 9*). The response to the stimulation of the interneuron group was quite stereotyped and highly reproducible for only a few of the hemilineages. This response consistency was confined to some of the hemilineage groups that evoked postural changes. We do not have detailed transmitter information for all of the hemilineage groups but we do know that the 9A and 5B clusters are GABA-immunopositive neurons (*Harris, 2012*). Therefore, the stereotypy resulting from their activation may be due to imposing inhibition at a particular level of sensory-motor integration. The hemilineages that evoked more complex behaviors typically showed a range of behaviors, although there was usually a dominant category of response that characterized that particular hemilineage group. Therefore, for these more complex behaviors, the response to hemilineage group stimulation was probabilistic rather than deterministic, in that the activity of these neurons enhanced the probability that a particular behavior would be performed. A similar phenomenon was seen in the behavioral responses of larval *Drosophila* to TRPA1 activation of different sparse sets of central neurons (*Vogelstein et al., 2014*). The larval behavioral responses also tended to be probabilistic in that a given set of neurons enhanced the probability of a behavioral response but their activity did not invariably lead to this response. Our discussion below focuses on the most frequently shown behaviors seen during the stimulation of each neuronal group.

*Figure 9* orders the tested hemilineages in terms of the increasing behavioral complexity seen as the neurons were stimulated. Responses ranged from simple postural changes, through rhythmic movements of walking or flight, to the complex behavioral sequence involved in takeoff. This diversity in behavior supports a hierarchical relationship amongst the various interneuron pools with some

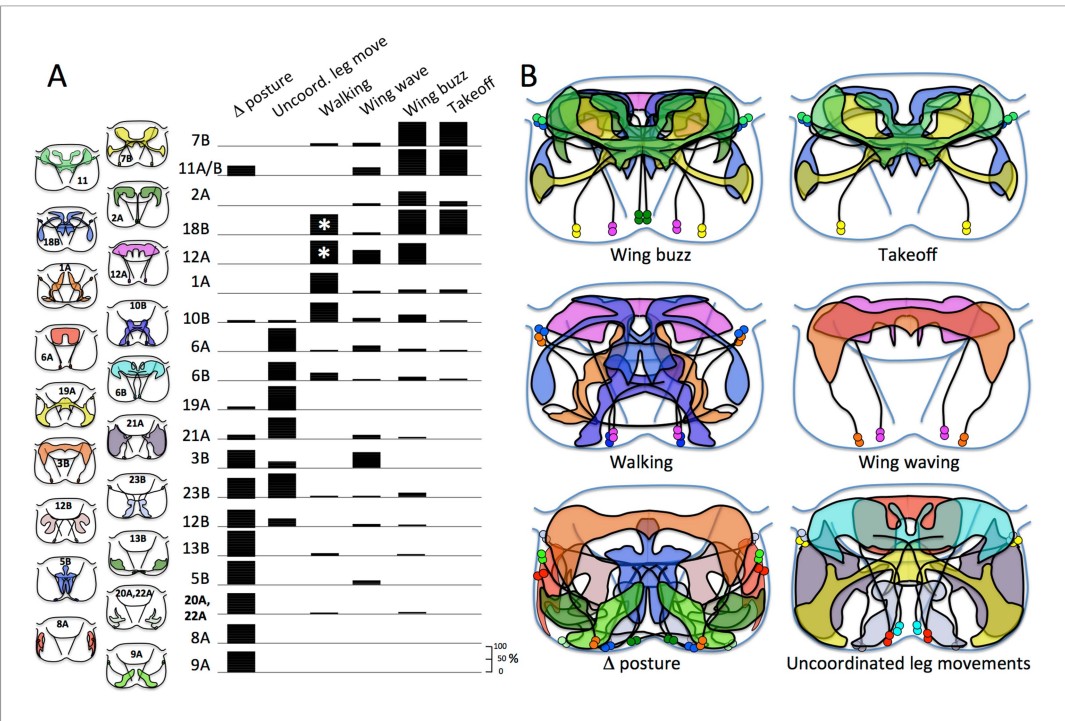

**Figure 9**. Relationship of hemilineages to neuron type and to classes of evoked behavior. (**A**) Summary of the range of behaviors in decapitated flies elicited by stimulation of the neurons in each of the hemilineage groups using TRPA1 expression and a heat ramp to activate the temperature-sensitive channel. Behaviors are divided into six categories explained in the text. Hemilineages are arranged according to the complexity of their behavioral responses. Most behavioral responses were sustained but a few hemilineages (*) showed a progression of behaviors during the heat ramp. Diagrams show the extent of the hemilineage's arbor in transverse views of the ventral nervous system at the level of the mesothorax (T2). The numbers of flies analyzed in each group ranged from 15 to 25. (**B**) Registration of the hemilineage arbors to a common outline and then overlapping the hemilineage groups in which at least 50% of the individuals showed the indicated types of behavior.

involved in the patterning of simple movements and others having a more 'upstream' function of assembling these simple movements into complex movements or sequences of behavior.

*Figure 9* also relates the projection patterns of the various hemilineages to the most common behavior seen when these neuronal groups are stimulated. Not surprisingly, the hemilineage groups that evoke changes in leg posture have their arbors concentrated in the leg neuropils. Most are confined to the leg neuropil except for the 3B interneurons, that have major projections into the tectulum and also evoke wing movements, and the 5B neurons, that are potentially GABAergic and may have a general suppressive function. Hemilineages that evoke coordinated or uncoordinated leg movements similarly have a strong leg neuropil component but their arbors also project into the medial neuropil and into the tectulum neuropil. Strong projections into the tectulum neuropil were also evident for all of the hemilineages that evoked wing movements of either low or high frequency. Two of the three hemilineages that evoked frequent takeoff had a strong projection into the T2 leg neuropil (the site of the jump motoneurons) in addition to their extensive arbor in the tectulum. Curiously, the third takeoff lineage, lineage 11A/B, lacked a T2 arbor but had substantial projections into the T1 and T3 leg neuropils. Overall the anatomy of the hemilineages conformed to the generally held idea that the leg neuropils deal with leg function while the tectulum neuropil is involved with wing-related behavior such as singing or flight. In addition, though, the tectulum and dorsomedial neuropil may be generally involved in complex, coordinated behavior in the thorax. Hemilineages, such as 1A and 10B, that evoke walking behavior but little in the way of wing movement nevertheless have arbor extension into the dorsomedial neuropil and tectulum, suggesting that these regions may be generally involved in the coordination of complex thoracic behavior regardless of whether it involves legs or wings.

Overall the hemilineages appear to have a modular function with cells in a given group being associated with particular behavioral responses. This notion is consistent with physiological data such as that from grasshoppers in which computational circuits are built from clusters of distinct types of interneurons (*Burrows, 1996*), at least some types of which have been shown to arise in the same lineage (*Shepherd and Laurent, 1992*). A similar strategy is seen in the vertebrate spinal cord in which different cell types arise from defined sets of precursors and the neurons from those cell types play specific roles in defined circuit motifs (*Grillner and Jessell, 2009*).

Our assessment of the functional roles of the hemilineages is obviously incomplete. We assayed hemilineage function only under one condition—decapitated flies standing at rest and showing occasional spontaneous grooming bouts. Stimulation of the hemilineages under other conditions, for example steady-state flight, will undoubtedly reveal more complexity in the behavioral functions of some of the hemilineages. A fuller understanding of lineage function will require a larger palate of behavioral tests.

A second caution comes from our demonstration that the vPR6 neurons, which are required for song production in the male (*von Philipsborn et al., 2011*), are part of hemilineage 12A cluster (*Figure 7*). During the course of stimulation of the 12A interneurons the decapitated flies showed a range of behaviors, some of which mimicked the movements displayed during courtship singing, but these soon degenerated into generalized wing vibration movements. We would expect that in the more complex dorsal hemilineages there might be specializations of function within the cluster and that the stimulation of all of the neurons at once might set up conflict states that might be hard to interpret. The specialization of the function of the vPR6 neurons suggest that some neurons within a hemilineage can be adapted for specialized behaviors that are, nevertheless, related to the basic function of the hemilineage as a whole.

## Hemilineages and the evolution of behavior

The segmental CNS of insects provides an unparalleled system in which to study the evolution and diversification of the nervous system. The advantage of the insect system lies in a stereotyped array of segmental NBs that has undergone little change through the course of insect evolution. Based on position, molecular markers and the types of early progeny that they produce, the homologous stem cells can be identified in insects as diverse as silverfish, grasshoppers and *Drosophila* (*Thomas et al., 1984*; *Truman and Ball, 1998*). Not only have the NBs been highly conserved, but their early neuronal progeny are also highly conserved as first elegantly shown by *Thomas et al. (1984)*. These early-born cells are the 'primary' neurons and their identities are determined by birth-order as encoded in

a transcription factor progression within the cycling NB (*Isshiki et al., 2001*). Many of these early-born cells serve a pioneer function, which explains the stereotyped pattern of tracts and commissures seen in the ventral neuropil throughout the insects (*Thomas et al., 1984*). The later-born 'secondary' neurons are born after the NB starts to express grainyhead (*Almeida and Bray, 2005*) and include late-born embryonic neurons as well as all of the neurons that are born during postembryonic life (see *Zhou et al., 2009*; for a discussion of primary vs secondary neurons). As summarized in *Figure 9*, the hemilineage units of these secondary neurons typically include neurons of related phenotypes that may participate in functionally related components of behavior. The interesting question is whether these cell classes are likewise conserved across the insects? Conservation in class character in widely different groups is well illustrated by the medial lineage. In grasshoppers the A hemilineage cells are GABAergic, engrailed-positive local interneurons and the B hemilineage cells are engrailed-negative, projection cells (*Jia and Siegler, 2002*), and the same is true in *Drosophila* (JW Truman, unpublished). Similarly, a comparative analysis of the clusters of GABA-immunoreactive neurons in the VNS of diverse insects (*Witten and Truman, 1998*) supports the contention that the properties of neurons in homologous hemilineages are highly conserved across much of the insects. Although one can homologize cell classes of secondary neurons across most of the insects, we think that it is unlikely that one could homologize individual neurons within a particular class.

This then brings us to the question of the conservation of function for the neurons of the various hemilineage classes. In grasshoppers ventral interneurons that run in the transverse tract receive input from wing afferents (*Watson and Burrows, 1983*) while in flies the 2A interneurons are the main neurons of the transverse tract and they can drive flight behavior. For the leg hemilineages, NB4-1 and NB3-1 in grasshopper make the medial and anteromedial spiking interneurons, respectively, of the leg circuit (*Burrows, 1996*), while in flies the homologous NBs (*Truman et al., 2004*) generate the 14A and 4B interneurons which are similar appearing local interneurons in the leg neuropil. Whether the functions of the neurons are exactly the same in the two species are as of yet unknown, but their involvement in leg function is clearly conserved. We expect that the functional roles of most of all of the hemilineages to be highly conserved through much of insect evolution. For the simpler hemilineages that contain units that work in parallel, addition or reductions in the number of neurons could enhance or reduce functionality by altering the spatial or temporal resolution of their level of the computational network. Another benefit of an increase in numbers of an interneuron type could be to provide a finer control over the speed of locomotion , as seen for the control of swimming speed in larval zebrafish (*McLean et al., 2008*) in which the increase in speed involves the recruitment of new interneurons and the repression of interneurons active at the slower speed. For higher level, more complex hemilineages, changes in neuronal properties/connectivity rather than numbers might be more important for generating behavioral changes.

Our data from *Drosophila* provide the first systematic analysis of the hemilineages that make up a major section of the insect CNS. For the thoracic segments it provides a reference to which information from other insects can be compared. It is important to remember, though, that the flight system of *Drosophila*, with its mesothoracic wings and metathoracic halteres, is a highly derived system compared with more basal insect groups. Also, in the fly about a third of the thoracic hemilineages die leaving few if any members (*Truman et al., 2010*). Consequently, some cell types that may have been important in the ancestral nervous system have been lost with the evolution of *Drosophila*'s derived pattern of flight. The other important caveat is that our analysis includes only the postembryonic secondary neurons, and so we are lacking access to the embryonic-born neurons, a small but extremely important component of the adult CNS.

Finally is the issue of whether the data from *Drosophila* have bearing on the functional organization of nervous systems outside of the arthropods, such as the vertebrate spinal cord. At this point in time we do not know. However, a thoracic unit of the fly thoracic nervous system has about 2500 pairs of neurons, most of which can be assigned to 33 hemilineage units. For many (most?) of the hemilineages one may be able to distill the functional essence of each unit down to one or two sets of cellular characteristics that may reflect the ancestral roles of these cells before the neuronal numbers were expanded. This would then provide us with a basic 'tool-kit' of neuron types that insects have used to construct their VNS. These features and the knowledge of the other units on which they preferably synapse may allow us to hypothesize an ancestral wiring diagram for a primitive insect/arthropod VNS. With such a wiring diagram in hand we would then be better able to look for functional homologies in a primitive vertebrate spinal cord such as the lamprey.

## Materials and methods

### Fly stocks

Flies were reared on standard cornmeal and molasses food at 25°C. All GAL4 lines are from the Rubin GAL4 collection (*Pfeiffer et al., 2008*; *Jenett et al., 2012*). Additional stocks used in this study: pJFRC180-20XUAS-IVS-Flp2::PEST in su(Hw)attP8, pJFRC180-20XUAS-IVS-Flp2::PEST in *attP40*; Actin5C-FRT>-dSTOP-FRT>-GAL4 in *attP18*; R24B02-GeneSwitch in *attP2*; pJFRC2-10XUAS-IVS-mCD8::GFP in *attP2* (*Pfeiffer et al., 2010*); R57C10-GAL80-6 in *su(Hw)attP8*; Actin5C-FRT>-dSTOP-FRT>-LexAp::65 in *su(Hw)attP5*; pJFRC19-13XLexAop2-IVS-myr::GFP in *attP40* (Pfeiffer et al., 2010); pJFRC177-10XUAS-FRT>-dSTOP-FRT>-myr::GFP in *attP18* (*Nern et al., 2011*); R57C10-LexA::p65 in *attP2* (*Pfeiffer et al., 2012*); pJFRC109-13XLexAop2-FRT>-dSTOP-FRT>-dTRPA1 in *VK00005*; pJFRC108-20XUAS-IVS-hPR::Flp-p10 in *VK00005*; pJFRC26-13XLexAop2-IVS-dTRPA1-WPRE in *attP40* and *attP2* (*Liu et al., 2012*); pJFRC40-13XLexAop2-FRT>-STOP-FRT>-myr::GFP in *attP40* and *su(Hw)attP5* (*Pfeiffer et al., 2010*; *Nern et al., 2011*); pJFRC67-3XUAS-IVS-Syt::GFP in *attP18* (*Zhang et al., 2002*; *Pfeiffer et al., 2010*; *Seelig and Jayaraman, 2013*); tsh-GAL80 (*Clyne and Miesenböck, 2008*).

### New genetic constructs

Standard molecular biology techniques, as previously described (*Pfeiffer et al., 2010*), were used in generating the vectors detailed in this study. *Drosophila* codon-optimized GeneSwitch (*Wang et al., 1994*, *1997*; *Burcin et al., 1998*; *Burcin et al., 1999*), a fusion of truncated human progesterone receptor ligand binding domain and the p65 activation domain, was synthesized by DNA2.0 (Menlo Park, CA). To generate pBPGeneSwitchUw, a 5′–*KpnI* to 3-*BamHI* fragment of codon-optimized GAL4 DNA-binding domain (*Pfeiffer et al., 2010*) and a 5′-*BamHI* to 3′-*HindIII* codon-optimized GeneSwitch fragment were cloned, as a triple ligation, into 5′-*KpnI* to 3′-*HindIII* digested pBPGAL4.2::p65Uw (*Pfeiffer et al., 2010*). GAL4, GAL80, LexA, and GeneSwitch drivers containing putative *cis-regulatory* modules from the Janelia Research Campus used in this study were generated as previously described (*Pfeiffer et al., 2008*). All Constructs were sequence verified.

pJFRC180-20XUAS-IVS-Flp2::PEST was generated by cloning a fusion of codon-optimized Flp2 and a C-terminal PEST sequence as a *5-XhoI* to 3′-*XbaI* fragment into a similarly digested pJFRC7-20XUAS-IVS-mCD8::GFP vector (*Pfeiffer et al., 2010*; *Nern et al., 2011*). pJFRC108-20XUAS-IVS-hPR::Flp-p10 (UAS-Flp-Switch) was generated as follows: Codon-optimized GeneSwitch was used as template to PCR the truncated hPR ligand binding domain as a 5′-*XhoI* to 3′-*KpnI* fragment. Codon-optimized Flp1 variant was amplified from pJFRC150-20XUAS-IVS-Flp1::PEST (*Nern et al., 2011*) as a *5′-KpnI* to 3′-*XbaI* fragment. The two fragments were digested and cloned, as a triple ligation, into 5′-*XhoI* to 3′-*XbaI* cut pJFRC82-20XUAS-IVS-Syn21-GFP-p10 (*Pfeiffer et al., 2012*).

Actin5C-FRT>-dSTOP-FRT>-GAL4 and Actin5C-FRT>-dSTOP-FRT>-LexAp::65 were generated in three steps: **First**, a 4290 bp fragment 5′ of the translation start site and encompassing the two promoters for *Drosophila act5C* (*Bond and Davidson, 1986*; *Chung and Keller, 1990*) were cloned in a sequential manner: The distal promoter of *act5C* was PCR amplified from *y; cn bw sp* genomic DNA (*Adams et al., 2000*) and cloned as a 5′-*FseI* to 3′-*KpnI*/*AgeI* into pBDPGAL4U (*Pfeiffer et al., 2008*). A second PCR was performed using *y; cn bw sp* genomic DNA to amplify a 5′-*AgeI* to 3′-*AgeI* fragment for the proximal promoter and cloned into the former construct yielding Actin5C-GAL4-hsp70T. **Second**, Actin5C-GAL4-hsp70T was then digested 5′-*KpnI* to 3′-*NotI* to replace the GAL4 with codon-optimized LexA::p65 (*Pfeiffer et al., 2010*) to generate Actin5C-LexA::p65-hsp70T. **Third**, both the Actin5C-GAL4 and–LexA::p65 constructs were digested with *KpnI* to insert a FRT flanked cassette consisting of hsp70 and SV40 transcriptional terminator sequences (*Nern et al., 2011*) to generate Actin5C-FRT>-dSTOP-FRT>-GAL4 and Actin5C-FRT>-dSTOP-FRT>-LexAp::65, respectively.

### RU486 treatments

GeneSwitch animals were treated with the progesterone mimic mifepristone (RU486, Sigma-Aldrich, St. Louis, MO). Either 1.5 or 7 mg RU486 was dissolved in 1.5 ml 70% ethanol and then mixed with 15 ml of melted fly food, which was then allowed to solidify and fed to larvae. For experiments in which

adults were fed RU486, 10–20 flies were placed in a food vial with RU486, allowed to feed for 3 days, then collected and dissected.

For surface application of RU486, parents were allowed to lay eggs in a food vial for a few days, then transferred to a fresh vial. Approximately 4 days after egg-laying, 60 µl of a ~10 mM RU486 stock solution (10 mg RU486 dissolved in 2 ml 95% ethanol) was applied to the surface of the food. At 24 hr after treatment, any larvae that had wandered and/or pupariated were discarded to ensure that test animals had fed on RU486 for at least 24 hr. At 48 hr after treatment, the subsequent wandering larvae and pupae (which had all fed on RU486 for 24–48 hr) were collected and transferred to an untreated food vial. These animals were then dissected at various times thereafter.

## Preparation and examination of tissues

Tissues were dissected in PBS (phosphate-buffered saline, pH 7.8, Cellgro by MediaTech, Inc., Manassas, VA) and fixed in 4% buffered formaldehyde overnight at 4°C. Fixed tissues were rinsed in PBS-TX (PBS with 1% Triton X-100, Sigma-Aldrich), then incubated overnight at 4°C in a cocktail of 10% normal donkey serum (Jackson ImmunoResearch, West Grove, PA), 1:1000 rabbit anti-GFP (Jackson ImmunoResearch), 1:40 rat anti-N-Cadherin (Developmental Studies Hybridoma Bank, Iowa City, IA), and 1:40 mouse anti-Neuroglian (Developmental Studies Hybridoma Bank). Tissues were then rinsed in PBS-TX and incubated overnight at 4°C with 1:500 AlexaFluor 488-conjugated donkey anti-rabbit, AlexaFluor 594-conjugated donkey anti-mouse, and AlexaFluor 649-conjugated donkey anti-rat (all from Invitrogen, Grand Island, NY). Tissues were then washed in PBS-TX, mounted onto poly-lysine-coated coverslips, dehydrated through an ethanol series, cleared in xylenes, and mounted in DPX mountant (Sigma–Aldrich). Nervous systems were imaged on a Zeiss LSM 510 confocal microscope at 40× with optical sections taken at 2 µm intervals. LSM files were contrast-enhanced as necessary and $z$-projected using ImageJ (http://rsbweb.nih.gov/ij/).

## Decapitation, neural activation, and video recordings

3 to 4 days after eclosion, adult females were lightly $CO_2$-anesthetized, sorted, and returned to food vials, where they were allowed to recover for 2 days. Females were then chilled in an iced vial, transferred to a cold plate (Teca, Chicago, IL) at 2°C, and quickly decapitated using microscissors in batches of 5–20. Flies were on the cold plate for less than 3 min. Decapitated flies were brushed back into a food vial and allowed to recover for at least 1 hr.

The heat-activated cation channel TRPA1 shows some activity at 25°C, and is fully active in the range of 27°C–32°C (*Hamada et al., 2008*). For TRPA1 activation, batches of 5–10 decapitated flies were tapped onto a sheet of paper, and any flies that were unable to right themselves were discarded. The paper was then placed on a hot plate (Teca) and ramped from 24°C to maximum activation temperature. For flies reared at 21°C, the maximum activation temperature was 32°C and the ramp took 45 s. For flies reared at 25°C, the maximum activation temperature was 37°C and the ramp took 55 s. Reference temperatures taken from the hot plate's internal sensor were manually marked in the videos.

For all lineages, low-speed (60 fps) videos were taken using a Dragonfly Express digital camera (Point Grey Research, Canada) controlled by the MATLAB Image Acquisition (ImAq) tool in the Image Acquisition Toolbox. Videos were taken from above, with a field of view diameter of approximately 50 mm. Recordings lasted 60 s, encompassing the entire temperature ramp and a period of time at maximum activation temperature.

For lineages that induced flight-related phenotypes, high-speed (1000–6000 fps) videos were taken with a Phantom v9 high-speed digital camera (Vision Research, Wayne, NJ) using the Phantom software. Flies were recorded individually or in groups of less than five, with a field of view approximately 5 mm in diameter. Recordings were saved from the camera's memory buffer from the period preceding a predetermined behavioral trigger-point, such as takeoff.

## Data analysis

Low-speed footage (60 fps) was played back using VirtualDub (http://virtualdub.org/) and/or ImageJ (http://rsbweb.nih.gov/ij/). The behavior of each fly during the heat ramp was annotated manually. High-speed footage was annotated manually using Phantom software. Behaviors were scored throughout the temperature ramp.

## Acknowledgements

We thank Lynn Riddiford, Minoro Koyama, Barry Dickson, David Shepherd and Wyatt Korff for comments and a critical reading of the manuscript, Todd Laverty, Karen Hibbard and the other FlyCore staff for help with fly crosses, and Alison Howard for administrative help. We also thank Tom Daniel of the University of Washington for early training of RMH prior to her transfer to Janelia Research Campus. Supported by HHMI.

## Additional information

### Funding

| Funder | Grant reference | Author |
| --- | --- | --- |
| Howard Hughes Medical Institute (HHMI) | Group Leader Budget | James W Truman |

The funder had no role in study design, data collection and interpretation, or the decision to submit the work for publication.

### Author contributions

RMH, Conception and design, Acquisition of data, Analysis and interpretation of data; BDP, JWT, Conception and design, Analysis and interpretation of data, Drafting or revising the article; GMR, Conception and design, Drafting or revising the article

## Additional files

### Supplementary file

• Supplementary file 1. Genotypes used for anatomical and behavioral characterization of the hemilineages and the amount of off-target expression in the ventral nervous system of each.

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
