## [Decision Letter]

Thank you for sending your work entitled “Neuron hemilineages are functional modules of the *Drosophila* Ventral Nervous System” for consideration at *eLife*. Your article has been favorably evaluated by K VijayRaghavan (Senior editor) and four reviewers, one of whom, Leslie Griffith, is a member of our Board of Reviewing Editors.

The Reviewing editor and the other reviewers discussed their comments before we reached this decision, and the Reviewing editor has assembled the following comments to help you prepare a revised submission:

While all the reviewers were very positive about this work, there was common concern about the level of description of both the anatomy and the behavioral phenotypes. This paper has the potential to be foundational for future studies of locomotor circuits, but to be useful to the field it must contain more information than it currently does. There are three areas that need to be expanded.

1) Anatomy. It is important to put the anatomy of the hemilineages, as visualized by the drivers, in context with the lineage labeled as a clone. These clones, and the pattern of neuroglian-positive tracts in the adult, have been described in the dissertation of the lead author, which is cited. It is difficult to compare the hemilineage projection patterns shown in this paper (which are presented as a single z-projection along the entire thickness of the VNS) with the clone data in Dr. Harris' thesis. More room should be given in the anatomy section of this paper to these aspects. The clones in relationship to hemilineages should be shown; logically they should encompass the two sister hemilineages labeled by drivers. If this is not always the case, it is an important fact to document.

2) Off-target cells. How was the “success of a genotype in generating a pure hemilineage” calculated? Were clones used as a comparison? The lack of information on clones (above) becomes an issue here. The only figure where the hemilineages are shown registered to a common template is the schematic Discussion Figure 8. This figure, and the raw data underlying it (i.e., registering the hemilineage patterns to a template) should be part of this paper. It might be good to show more clearly how the current hemilineage labeling strategies compare to MARCM labeled clones with regard to targeting Gal4/LexA activity to entire hemilineages. This would also address the question of off-target cells. The aggregate information (with numbers for each line, not simply descriptors) could be presented in table format.

3) Behavior. The behavioral phenotypes need to be documented more precisely in Figures 4 and 5. In many cases the reader cannot extract the author's conclusions from viewing the figures. For example, there is no figure panel that convincingly shows flies lacking interlimb coordination – one of their conclusions. High speed recordings with frames documenting each movement would provide the required information, perhaps with a simplified stick figure summary of the leg/joint movements. It would also be interesting to discuss the fact that all hemilineages generated motor output. Are there no inhibitory neurons in the thorax? Are they embedded within excitatory populations in each hemilineage?

---

## [Author Response]

*1) Anatomy. It is important to put the anatomy of the hemilineages, as visualized by the drivers, in context with the lineage labeled as a clone. These clones, and the pattern of neuroglian-positive tracts in the adult, have been described in the dissertation of the lead author, which is cited. It is difficult to compare the hemilineage projection patterns shown in this paper (which are presented as a single z-projection along the entire thickness of the VNS) with the clone data in Dr. Harris' thesis. More room should be given in the anatomy section of this paper to these aspects. The clones in relationship to hemilineages should be shown; logically they should encompass the two sister hemilineages labeled by drivers. If this is not always the case, it is an important fact to document*.

We have expanded the anatomical aspects of the paper with a more detailed dorsal view and cross-section projections of each hemilineage. In terms of MARCM clones, a few have already been published (Brown and Truman) and we have added a few supplemental figures of MARCM clones for some of the complex lineages. A comprehensive treatment showing MARCM clones for all of the lineages and how they vary in each of the three thoracic segments is being prepared (Shepherd, Sahota, Court, Harris, Truman and Williams, in preparation) but adding this amount of data would be beyond the scope of this paper.

*2) Off-target cells. How was the* “*success of a genotype in generating a pure hemilineage*” *calculated? Were clones used as a comparison? The lack of information on clones (above) becomes an issue here. The only figure where the hemilineages are shown registered to a common template is the schematic Discussion*
Figure 8*. This figure, and the raw data underlying it (i.e., registering the hemilineage patterns to a template) should be part of this paper. It might be good to show more clearly how the current hemilineage labeling strategies compare to MARCM labeled clones with regard to targeting Gal4/LexA activity to entire hemilineages. This would also address the question of off-target cells. The aggregate information (with numbers for each line, not simply descriptors) could be presented in table format*.

The clone paper cited above includes registering the clones onto a common template. Although the MARCM clone analysis was started by Dr. Harris, it was substantially expanded and carried to completing after her death. Consequently, we think it should be a separate paper. For the purpose of the present paper, our first concern was that we were getting the members of one hemilineage but none of the members of its sister hemilineage. We met this goal in all cases. Supplemental file 1 documents the amount of non-hemilineage contamination for each line. We accepted lines that had some scattered weak expressing cells in the thorax. We were more tolerant of accepting expression in abdominal interneurons. Lines with prominent sensory expression were not used. As described in the text, we also controlled for the effects of descending axons when such were present.

*3) Behavior. The behavioral phenotypes need to be documented more precisely in*
Figures 4 and 5*. In many cases the reader cannot extract the author's conclusions from viewing the figures. For example, there is no figure panel that convincingly shows flies lacking interlimb coordination – one of their conclusions. High speed recordings with frames documenting each movement would provide the required information, perhaps with a simplified stick figure summary of the leg/joint movements. It would also be interesting to discuss the fact that all hemilineages generated motor output. Are there no inhibitory neurons in the thorax? Are they embedded within excitatory populations in each hemilineage*?

Details of the behavioral responses have been expanded with a description for each hemilineage and a supplementary video showing the behavioral responses of the decapitated flies to the heat ramp. We only have fragmentary transmitter data for the hemilineages at present. In some cases, though, such as for hemilineage 12B, we know that all of the members are GABA immunoreactive. Stimulating these cells results in tonic leg extension.